# Analysis of 72,469 UK Biobank exomes links rare variants to male-pattern hair loss

**Sabrina Katrin Henne** [1], **Rana Aldisi** [2], **Sugirthan Sivalingam**[2,3], **Lara Maleen Hochfeld**[1], **Oleg Borisov** [2], **Peter Michael Krawitz**[2], **Carlo Maj** [2,4], **Markus Maria Nöthen** [1] & **Stefanie Heilmann-Heimbach** [1] ✉

Male-pattern hair loss (MPHL) is common and highly heritable. While genome-wide association studies (GWAS) have generated insights into the contribution of common variants to MPHL etiology, the relevance of rare variants remains unclear. To determine the contribution of rare variants to MPHL etiology, we perform gene-based and single-variant analyses in exome-sequencing data from 72,469 male UK Biobank participants. While our population-level risk prediction suggests that rare variants make only a minor contribution to general MPHL risk, our rare variant collapsing tests identified a total of five significant gene associations. These findings provide additional evidence for previously implicated genes (*EDA2R*, *WNT10A*) and highlight novel risk genes at and beyond GWAS loci (*HEPH*, *CEPT1*, *EIF3F*). Furthermore, MPHL-associated genes are enriched for genes considered causal for monogenic trichoses. Together, our findings broaden the MPHL-associated allelic spectrum and provide insights into MPHL pathobiology and a shared basis with monogenic hair loss disorders.

Male-pattern hair loss (MPHL), or androgenetic alopecia, is the most common form of hair loss, with a lifetime prevalence of ~80% in European men. MPHL is characterized by progressive and androgen-dependent hair loss in the frontotemporal region and vertex of the scalp[1]. Affected men may experience psychosocial effects[2], and lack well-tolerated and effective treatment options[3,4].

Early twin studies estimated that ~80% of the observed phenotypic variance of MPHL is attributable to genetic factors[5,6]. Subsequent genome-wide association studies (GWAS) have yielded substantial insights into the genetic basis of MPHL via the identification of more than 600 independent genetic risk variants at more than 350 genomic loci, which together explain ~39% of the phenotypic variance[7–17].While these data have highlighted a number of plausible candidate genes and pathways, the majority of GWAS risk variants are common variants (minor allele frequency (MAF) ≫ 1%) located in non-coding areas of the genome, which renders pinpointing of disease mechanisms and causal genes notoriously difficult.

In contrast, fewer data are available concerning the potential contribution to MPHL etiology of rare variants (MAF < 1%). A previous study on MPHL, which analyzed imputed genotyping data from the UK Biobank (UKB), estimated that the contribution of rare variants (MAF 0.0015% − 1%) to MPHL heritability was close to 0%[13]. However, imputed genotyping data do not offer comprehensive insights on rare variants, the systematic study of which has been hampered by the limited availability of whole genome or in the context of (rare) coding variants, whole exome sequencing (WES) data from adequately sized cohorts. Since 2019, the analysis of (rare) variants in coding areas of the genome has been facilitated by the availability of a large WES data set created by UKB[18,19]. The UKB resource further contains self-report data on MPHL, thereby for the first time enabling the investigation of a potential relevance of rare variants to MPHL pathogenesis.

The aim of the present study therefore was to perform the first exome-based analysis on MPHL in a tranche of 200,629 exomes from

[1]Institute of Human Genetics, University of Bonn, School of Medicine & University Hospital Bonn, Bonn, Germany. [2]Institute for Genomic Statistics and Bioinformatics, University of Bonn, Bonn, Germany. [3]Department of Medical Biometry, Informatics and Epidemiology, University of Bonn, Bonn, Germany. [4]Center for Human Genetics, University Hospital of Marburg, Marburg, Germany. ✉e-mail: sheilman@uni-bonn.de

the UKB. Gene-based analyses (SKAT-O and GenRisk) and single-variant tests were used to investigate whether rare variants showed association with MPHL in a final set of 72,469 men. To interpret the association findings, multiple follow-up analyses were performed. A schematic overview of the study workflow is depicted in Fig. 1. Our first systematic analysis of the contribution of rare variants to MPHL etiology broadens the allelic spectrum of previously reported candidate genes (*EDA2R*, *WNT10A*), yields evidence for novel MPHL candidate genes both at and beyond known GWAS loci (*HEPH*, *CEPT1*, *EIF3F*), suggests an association between genotrichoses and the common MPHL phenotype and provides a basis for future investigations of the contribution of rare variants to MPHL pathobiology.

## Results

### Data set characteristics
After quality control, the final data set comprised the data of 72,469 men aged 39–82 years. Our continuous model, all-model and two-as-control model comprised 72,024 unrelated (kinship < 0.0442) men. Of these, 49,640 with any signs of baldness (pattern 2–4) were classified as cases (case-control ratio 2.2:1) in the all-model, and 33,454 (pattern 3 or 4) were classified as cases in the two-as-control model (case-control ratio 1:1.2). The age distribution per MPHL pattern group is shown in Fig. 2. The extreme model comprised 17,053 unrelated men, of whom 6523 relatively younger men (age < 60 years) with significant balding (pattern 4) were classified as cases and 10,530 elderly men (age ≥ 60 years) with no signs of balding were classified as controls (case-control ratio 1:1.6).

After filtering for per-sample and per-individual missing rates (<5%) and Hardy-Weinberg-Equilibrium ($P_{HWE} > 10^{-6}$), a total of 2,656,761 rare (MAF < 1%), nonsynonymous variants in 18,946 protein-coding genes remained for analysis in the SKAT-O and single-variant association tests (Fig. 1), with 239,082 variants in 18,449 genes meeting the more stringent high impact threshold (frameshift, splice acceptor-, splice donor-, and start- or stop-altering variants, transcript ablations and transcript amplifications). For the GenRisk analyses, a total of 16,211,028 rare (MAF < 1%) variants in 18,848 genes remained after filtering.

Analyses were performed to assess the optimal number of top principal components (PCs) to correct for. In the association tests of imputed genotype data with a variable number of included top PCs, minimum genomic inflation factor values were generated when including 14–20 PCs in the continuous model, 14–15 PCs in the all-model, 14–19 PCs in the two-as-control model, and 5 PCs in the extreme model (see Supplementary Fig. 1). Based on these findings, we opted to correct for 14 PCs in the continuous-, all- and two-as-control models, and for 5 PCs in the extreme model.

### Single-variant association analyses
In a first step, we tested for an association of individual rare coding variants to MPHL. The analyses identified two genome-wide significant variants ($P < 8 \times 10^{-9}$) in the continuous- and all-model (Fig. 3, Supplementary Fig. 2, Supplementary Data 1). The two genome-wide significant variants, i.e., 23:66604439:G:A (rs12837393, MAF = $5.5 \times 10^{-3}$, $P_{continuous} = 3.0 \times 10^{-12}$, beta$_{continuous}$ = 0.19, $P_{all} = 4.8 \times 10^{-10}$, odds-

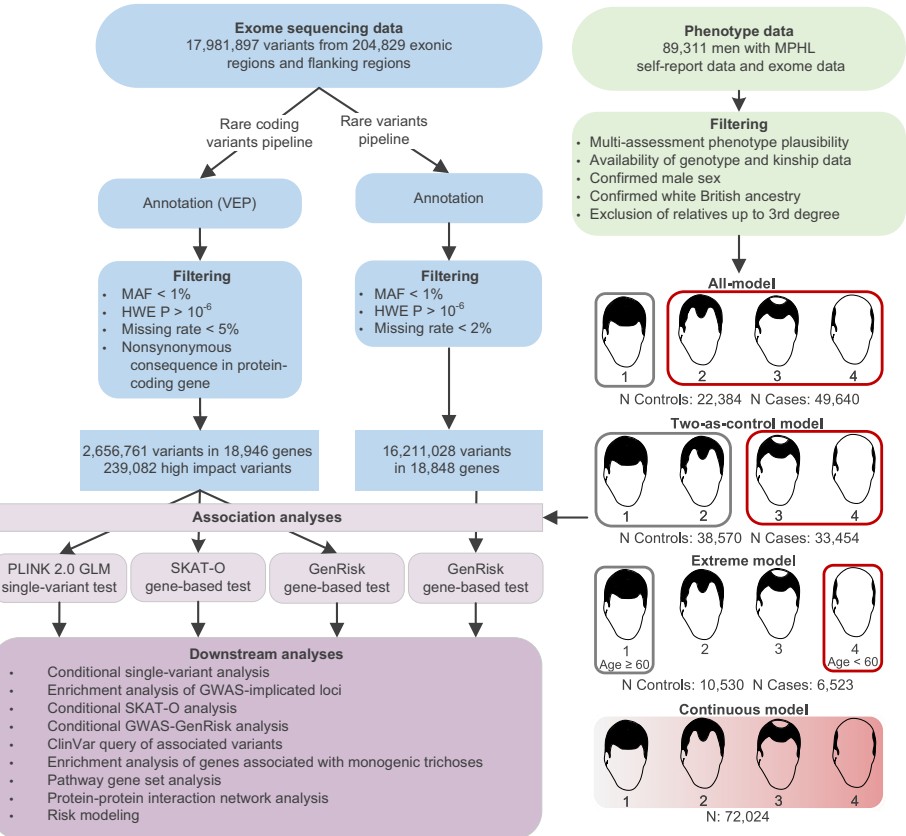

**Fig. 1 | Overview of the analysis workflow.** Exome and phenotype data obtained from the UKB were processed and used in three types of association analysis: GenRisk, SKAT-O, and single-variant testing. Four different phenotype models were used, of which three distinguishing cases (red) and controls (grey), as well as one continuous phenotype model. To interpret the association findings, several downstream follow-up analyses were performed. VEP ensembl variant effect predictor, HWE Hardy-Weinberg-equilibrium, MAF minor allele frequency, GWAS genome-wide association study, MPHL male-pattern hair loss. MPHL pattern diagrams adapted from the UK Biobank survey accessible at https://biobank.ctsu.ox.ac.uk/crystal/refer.cgi?id=100423 and reproduced by kind permission of UK Biobank ©.

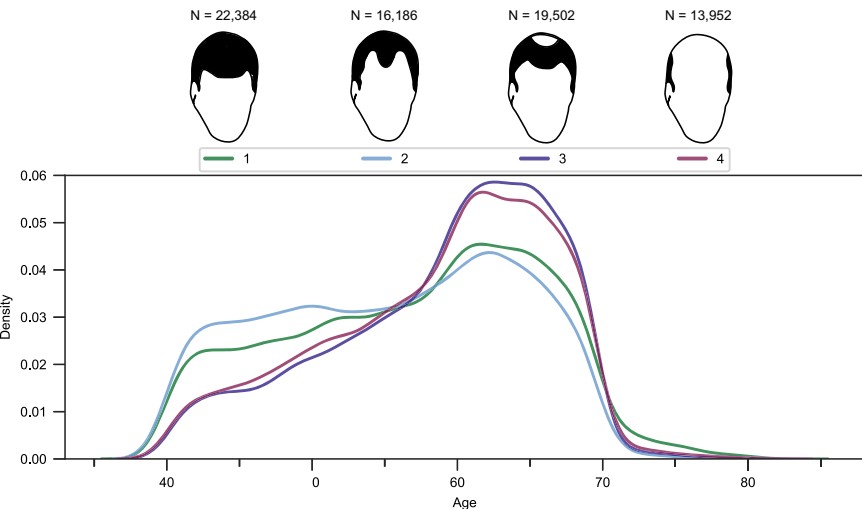

**Fig. 2 | Phenotypic distribution within the final set of 72,024 men in the continuous-, all- and two-as-control model.** Density plot showing the age distribution per male-pattern hair loss (MPHL) pattern group. The number of individuals in each MPHL pattern group is shown above the plot. MPHL pattern diagrams adapted from the UK Biobank survey accessible at https://biobank.ctsu.ox.ac.uk/crystal/refer.cgi?id=100423 and reproduced by kind permission of UK Biobank ©.

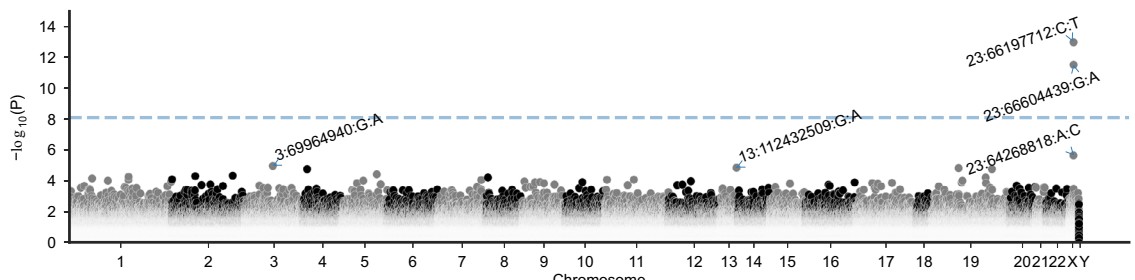

**Fig. 3 | Results of the single-variant analysis for the continuous model.** Manhattan plot of single-variant association results for the continuous model. The dashed line denotes the selected genome-wide threshold for multiple testing in single-variant tests ($8 \times 10^{-9}$). The y-axis depicts $-\log_{10}(P)$ of the unadjusted $P$-value obtained from linear regression (two-sided). The top 5 variants were annotated.

ratio $[OR]_{all} = 1.53$, $r^2_{sentinel\ SNP} = 1.6 \times 10^{-4}$, $D'_{sentinel\ SNP} = 0.35$) and 23:66197712:C:T (rs151003259, MAF $= 2.0 \times 10^{-3}$, $P_{continuous} = 1.0 \times 10^{-13}$, $beta_{continuous} = -0.35$, $P_{all} = 2.9 \times 10^{-10}$, $OR_{all} = 0.59$, $r^2_{sentinel\ SNP} = 4.5 \times 10^{-7}$, $D'_{sentinel\ SNP} = 1.0$) (GRCh38), are missense variants located within *EDA2R* and *HEPH* respectively. Notably, the T allele of 23:66197712:C:T was exclusively observed in combination with the MPHL risk allele (MAF > 0.99) of the respective GWAS sentinel SNP.

To assess whether the observed single-variant associations were independent of common variant associations previously identified through GWAS, all single-variant analyses were repeated with conditioning for 622 lead SNPs previously implicated in a UKB-based GWAS on MPHL[13] (Supplementary Fig. 3, Supplementary Data 1). Neither of the previously significant single variants retained genome-wide significance after conditioning. While an association signal was retained for the variant 23:66604439:G:A in *EDA2R* ($P_{all} = 4.0 \times 10^{-4}$), the 23:66197712:C:T variant in *HEPH* was not independent of the GWAS lead SNPs ($P_{all} = 0.35$). Several variants retained a relatively low $P$-value even after conditioning, indicating a strong association that was independent from common GWAS variants. For instance, among the top ten variants post-conditioning were 3:69964940:G:A (rs149617956, located in *MITF*, $P_{continuous} = 5.4 \times 10^{-6}$), 2:218882368:C:A (rs121908119, located in *WNT10A*, $P_{two-as-control} = 6.1 \times 10^{-6}$), 21:44499878:C:T (rs138480801, located in *TSPEAR*, $P_{two-as-control} = 6.9 \times 10^{-6}$), 11:46366461:G:T (rs901998, located in *DGKZ*, $P_{two-as-control} = 1.1 \times 10^{-5}$), and 23:67711453:C:A (rs1800053, located in *AR*, $P_{continuous} = 1.8 \times 10^{-5}$).

### Gene-based association analyses

To assess the cumulative contribution of rare variants to MPHL, we performed gene-based association analyses using SKAT-O[20] and GenRisk[21], a new burden association test which upweights rarer and more deleterious variants (based on CADD). We applied the GenRisk test to a data set of both coding and non-coding rare variants, as well as to coding rare variants identical to the variant set used in the SKAT-O analysis. The SKAT-O analysis based on 2,656,761 variants from all ten variant consequence categories identified two genes with a genome-wide significant association ($P < 2.6 \times 10^{-6}$) to MPHL: *EDA2R* ($P_{continuous} = 1.4 \times 10^{-8}$); and *HEPH* ($P_{continuous} = 7.3 \times 10^{-9}$) (Fig. 4, Supplementary Data 2). No significantly associated genes were identified based on SKAT-O analyses of high-impact variants, with the top association, *WNT10A*, yielding a $P$-value of $7.8 \times 10^{-6}$ in the two-as-control model (Supplementary Fig. 4, Supplementary Data 2).

The GenRisk analyses identified a total of three significantly associated genes ($P < 2.6 \times 10^{-6}$) across the four phenotype models: *EDA2R* ($P_{continuous} = 1.8 \times 10^{-6}$), *CEPT1* ($P_{all-model} = 2.1 \times 10^{-6}$), and *WNT10A* ($P_{two-as-control} = 2.2 \times 10^{-6}$) (Fig. 5, Supplementary Data 3). The *CEPT1* association finding is likely attributable to a combination of coding and non-coding variants with high CADD scores, and mainly driven by the MPHL pattern groups 2 and 3. Whether this reflects a biological aspect has to be determined by further analyses. The GenRisk analyses based on only coding variants further identified a significant association with *EIF3F* ($P_{two-as-control} = 2.5 \times 10^{-6}$) (Fig. 6, Supplementary Data 3).

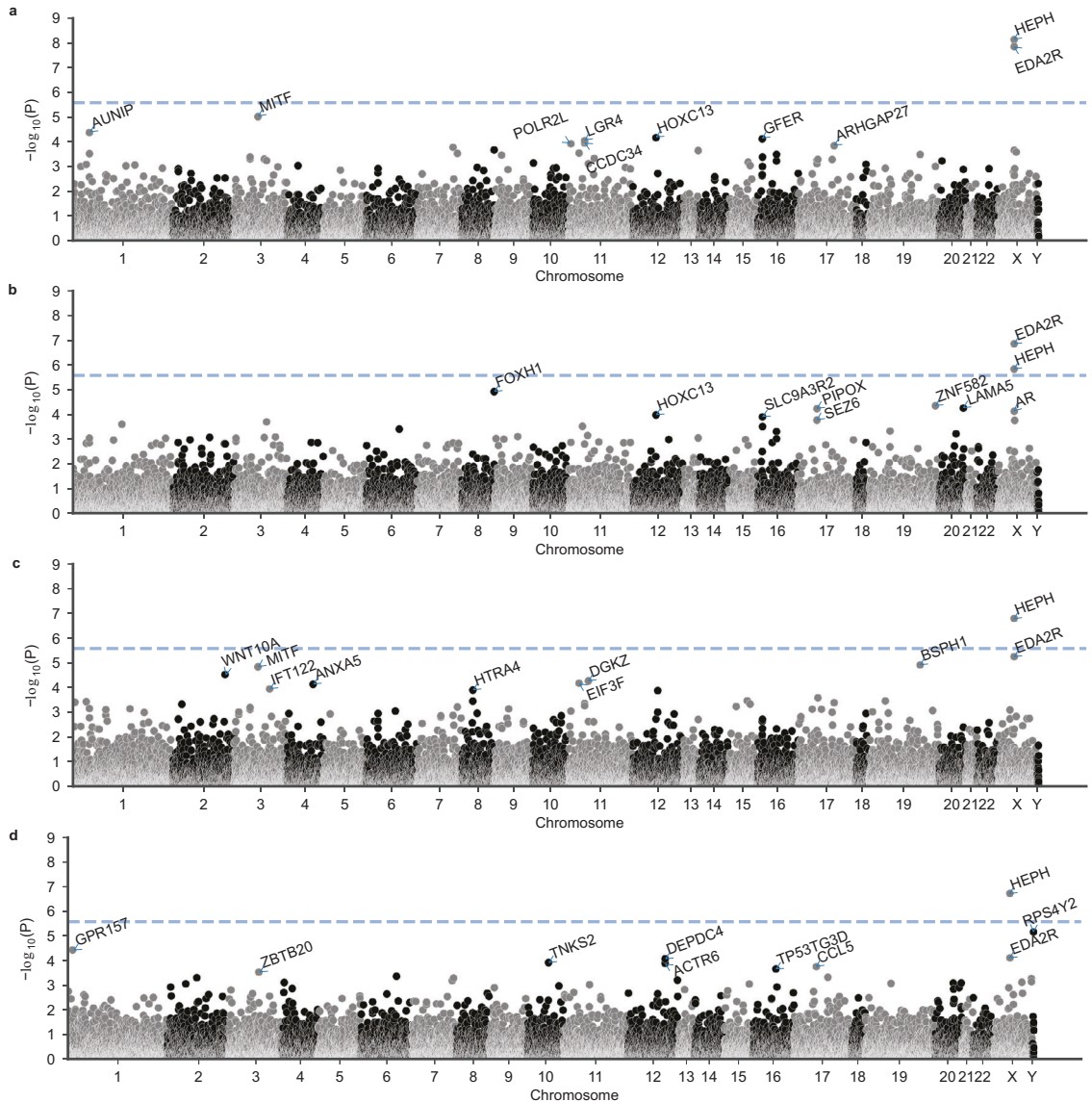

**Fig. 4 | Results of the SKAT-O gene-based analysis.** Results are shown for (**a**) the continuous model; (**b**) the all-model; (**c**) the two-as-control model; and (**d**) the extreme model. The y-axes depict $-\log_{10}(P)$ of the unadjusted P-value. The dashed line denotes the Bonferroni threshold for multiple testing in the gene-based analyses ($2.6 \times 10^{-6}$). The top 10 genes per analysis were annotated.

Comparison with an in-house data set on human hair follicle expression[22] revealed that all five MPHL-associated genes (*EDA2R*, *HEPH*, *CEPT1*, *WNT10A*, *EIF3F*) are expressed in human hair follicles. Of these, *EDA2R*, *HEPH* and *WNT10A* are located at previously implicated MPHL-GWAS risk loci. An enrichment of a less stringent set of gene associations ($P < 3 \times 10^{-3}$ in the SKAT-O or GenRisk analyses) was observed in regions ±1 Mb of published MPHL-GWAS lead SNPs ($P = 5.6 \times 10^{-15}$, overlap 192/595 genes). This was supported by the FUMA GENE2FUNC analysis, which identified an enrichment of these gene associations and MPHL GWAS findings reported in the GWAS catalog.

Conditional SKAT-O analyses were performed in order to determine whether the significant associations findings for *EDA2R* and *HEPH* were driven by the genome-wide significantly associated variants 23:66604439:G:A and 23:66197712:C:T, respectively. The P-values of *HEPH* and *EDA2R* both before and after the exclusion of these two variants are shown in Table 1. Notably, the association with *EDA2R* appears to have been driven very strongly by 23:66604439:G:A. In contrast, the effect of 23:66197712:C:T seems to have been less pronounced, since the conditional analyses for *HEPH* generated low P-values (albeit non genome-wide significant), particularly in the two-as-control and the extreme model.

**Conditional GWAS-GenRisk analysis**
A conditional GWAS-GenRisk analysis was performed to test whether common variants implicated by GWAS are independent from GenRisk gene scores (Supplementary Data 4). The distribution of the differences in $-\log_{10}(P)$ with and without GenRisk gene score correction is shown in Supplementary Fig. 5. These data indicate no systematic dependence between common variants implicated by GWAS and GenRisk gene scores, as a large majority of tested common variants (99.89%) are not or only minimally impacted ($|\Delta-\log_{10}(P)| < 1$) by correction for any gene score. However, the GenRisk scores of the genes *EDA2R* and *WNT10A* show some attenuation of the common variant GWAS signal at their respective loci. In contrast, the associated gene *HEPH* and e.g., the *AR* gene do not show any such attenuation (Supplementary Fig. 6).

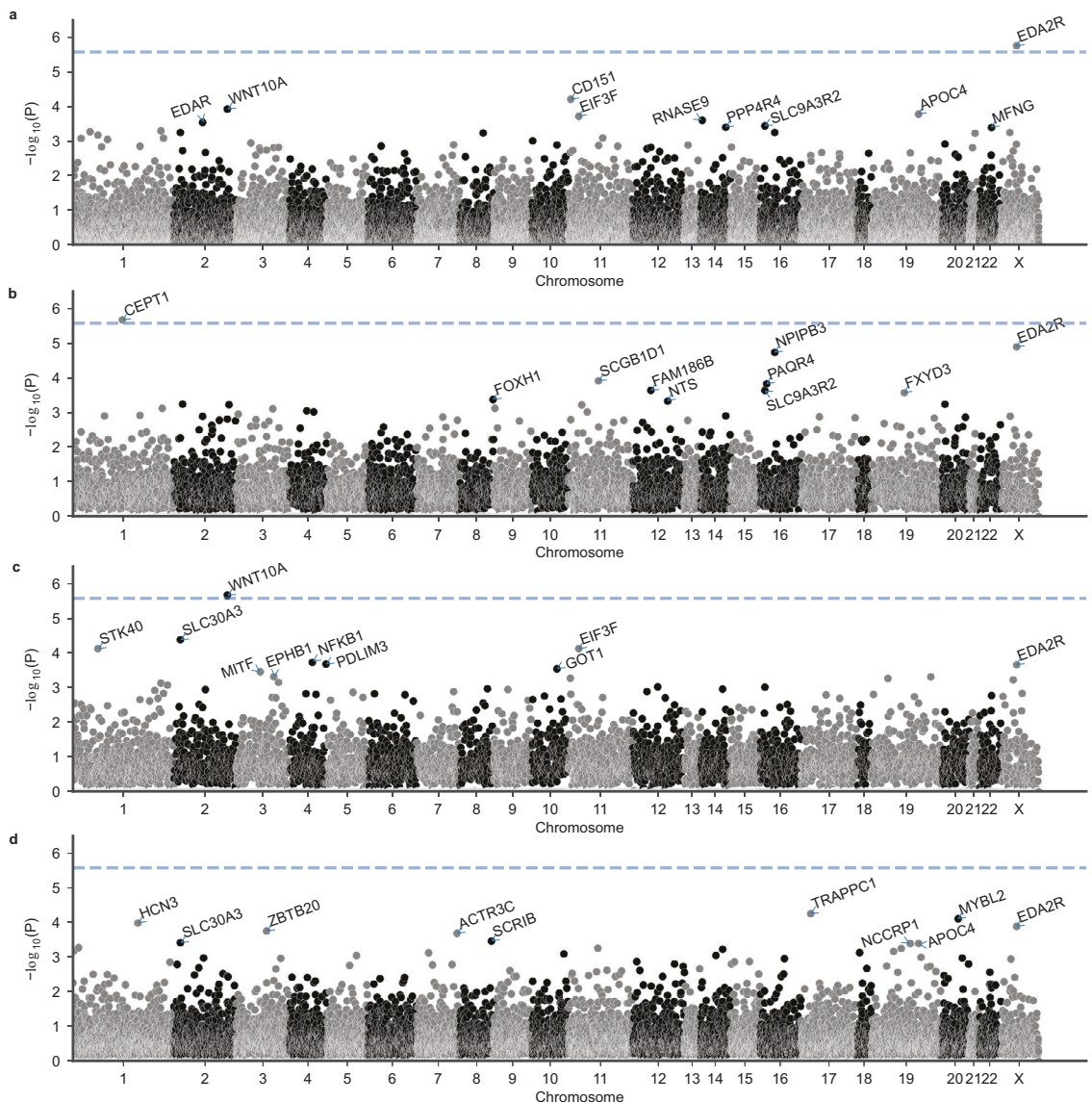

**Fig. 5 | Results of the GenRisk gene-based analysis.** Results are shown for (**a**) the continuous model; (**b**) the all-model; (**c**) the two-as-control model; and (**d**) the extreme model. The *y*-axes depict $-\log_{10}(P)$ of the unadjusted *P*-value. The dashed line denotes the Bonferroni threshold for multiple testing in the gene-based analyses ($2.6 \times 10^{-6}$). The top 10 genes per analysis were annotated.

## Overlap with genotrichoses

The inspection of ClinVar (Supplementary Data 5) revealed that MPHL-associated variants comprise several variants that have been reported as pathogenic for monogenic trichoses. A systematic enrichment analysis of genotrichosis-associated genes[23–27] amongst a less stringent set of gene associations ($P < 3 \times 10^{-3}$ in the SKAT-O or GenRisk analyses) revealed a significant enrichment ($P = 1.1 \times 10^{-4}$). The total overlap across all association analyses comprised the genes *WNT10A*, *HOXC13*, *DSP*, *LPAR6*, *ALX4*, *EDAR*, *CDH3*, *HR*, and *SPINK5*. Notably, two of the top associated single variants (albeit not genome-wide significant), i.e., 2:218882368:C:A ($P_{\text{two-as-control}} = 4.1 \times 10^{-5}$) and 21:44499878:C:T ($P_{\text{two-as-control}} = 9.0 \times 10^{-6}$), which are located in *WNT10A* and *TSPEAR* respectively, were reported to be pathogenic for ectodermal dysplasia in previous studies[28,29].

## Pathway gene set and network analyses

Pathway-based gene set enrichment analysis of a less stringent set of 559 MPHL-associated genes ($P < 3 \times 10^{-3}$ in either the SKAT-O or the GenRisk analyses) revealed an enrichment of MPHL-associated genes in TGF-beta signaling (false discovery rate [FDR] = 0.040) and SMAD2/ 3:SMAD4 transcriptional regulation (FDR = 0.021) (Supplementary Data 6). A protein-protein interaction network analysis of a less stringent set of 86 MPHL-associated genes ($P < 3 \times 10^{-4}$ in the SKAT-O or the GenRisk analyses) detected enrichments with ectodermal dysplasia genes (FDR = $2.6 \times 10^{-3}$, overlapping genes *EDA2R*, *WNT10A*, *EDAR*, *HOXC13* and *IFT122*) and genes assigned to the gene ontology term hair follicle development (FDR = 0.014, overlapping genes *WNT10A*, *EDAR*, *LAMA5*, *HOXC13*, *LGR4* and *ALX4*) (Supplementary Fig. 7).

## Risk modeling

To evaluate the contribution of rare variants to MPHL, a risk prediction model integrating MPHL polygenic risk scores (PRS) and GenRisk gene-based scores was created (Fig. 7), as based on rare variants (MAF < 1%), age, sequencing batch and top PCs. The PRS-only risk model achieved medium discriminative power similar to the MPHL PRS model previously published by Hagenaars et al.[14] in distinguishing no hair loss (pattern 1) from severe hair loss (pattern 4), at least moderate hair loss (pattern 3–4) and at least slight hair loss (pattern 2–4), as measured by the area under the curve (AUC) ($\text{AUC}_{\text{severe}} = 0.791$, $\text{AUC}_{\text{moderate}} = 0.732$, $\text{AUC}_{\text{slight}} = 0.693$) when

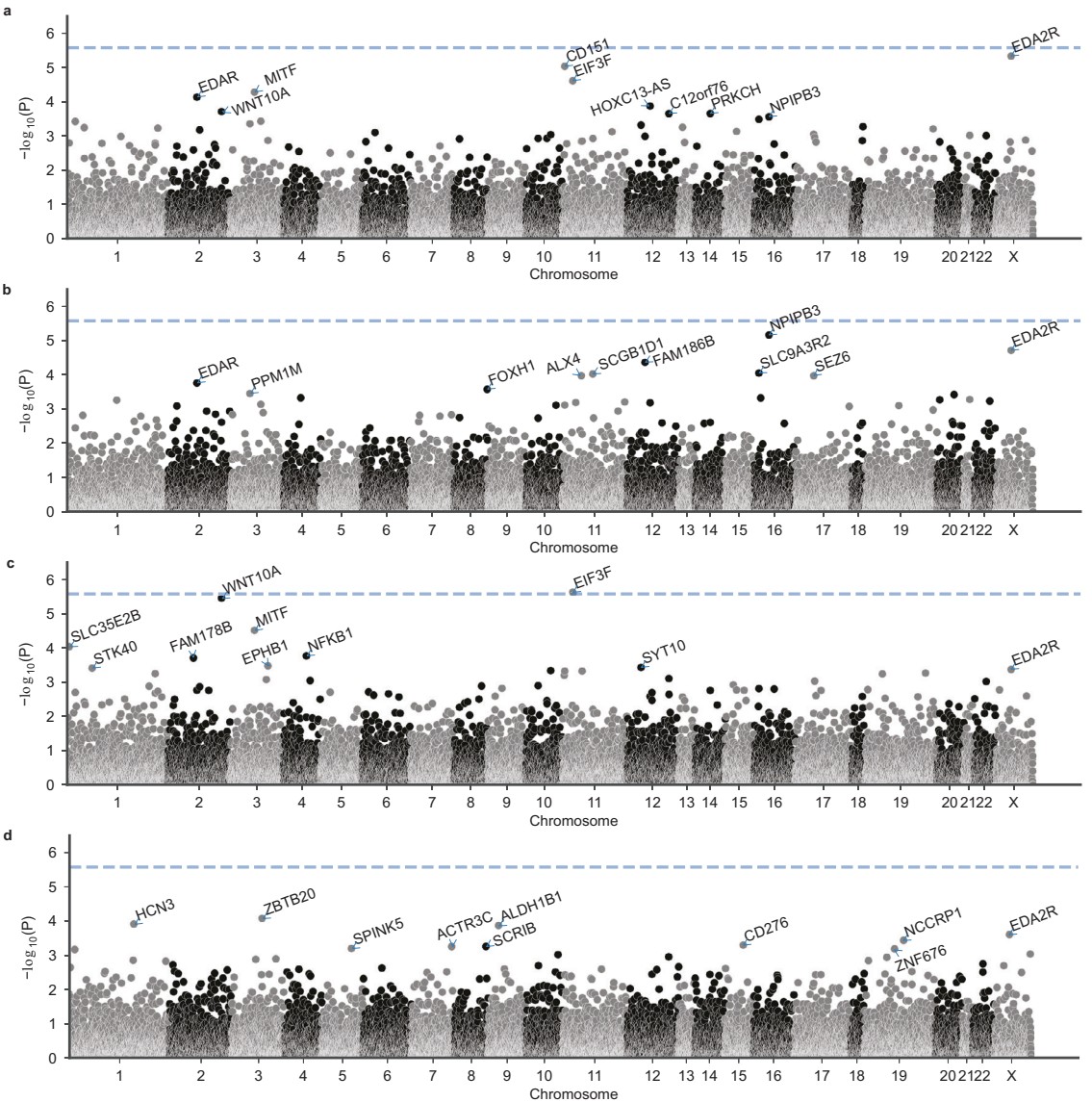

**Fig. 6 | Results of the GenRisk gene-based analysis based on only coding variants.** Results are shown for (**a**) the continuous model; (**b**) the all-model; (**c**) the two-as-control model; and (**d**) the extreme model. The y-axes depict $-\log_{10}(P)$ of the unadjusted P-value. The dashed line denotes the Bonferroni threshold for multiple testing in the gene-based analyses ($2.6 \times 10^{-6}$). The top 10 genes per analysis were annotated.

considering the full cohort of 72,024 males. In the test data set, the PRS-only model yielded slightly lower predictive power ($AUC_{severe} = 0.725$, $AUC_{moderate} = 0.687$, $AUC_{slight} = 0.647$). A risk model based exclusively on the gene-based risk score, which integrated all gene-based scores into one, showed low discriminative power ($AUC_{severe} = 0.560$, $AUC_{moderate} = 0.557$, $AUC_{slight} = 0.508$). Integration of PRS and gene-based risk scores generated only minimal to no increase in discriminative power compared to the PRS-only model ($AUC_{severe} = 0.726$, $AUC_{moderate} = 0.686$, $AUC_{slight} = 0.646$). Despite the high number of associated genes, this largely confirms earlier observations that rare variants explain only a minor fraction of the genetic risk for MPHL at population-level[13].

## Discussion

MPHL is a complex, common trait for which a large number of risk loci and variants have already been characterized via analyses of common variation[7–17]. The main aim of the present study was to analyze the extent to which rare variants contribute to MPHL. A previous study of MPHL, which was based on imputed genotyping data from the UKB, showed that the contribution of rare variants (MAF between 0.0015%

and 1%) to MPHL heritability was close to 0%[13]. To reassess this finding, we accessed a large exome sequencing data set from the UKB in order to perform a systematic analysis of rare variants in coding areas of the genome.

In line with previous reports that suggest a minor contribution of rare variants to MPHL heritability, our risk prediction models showed that the inclusion of gene-based scores that are based on rare variants into existing risk prediction models based on common variants made little to no contribution to discriminative power between cases and controls. This is also reflected in the low number of significant association findings in our single-variant analysis. Both rare variant associations identified ($P < 8 \times 10^{-9}$) have already been reported at genome-wide significance in GWAS[13].

The SKAT-O and GenRisk gene-based analyses detected significant associations with rare variants in five genes ($P < 2.6 \times 10^{-6}$), which, while limited, offers important insights into MPHL biology, and may be etiologically relevant for individual risk. The identified gene associations comprise both previously implicated and novel MPHL candidate genes. Genes previously implicated by GWAS include *EDA2R* (ectodysplasin A2 receptor), one of the flanking genes at the most

strongly associated MPHL GWAS locus on chromosome (chr) X (*AR/EDA2R* locus)[30] and *WNT10A* (Wnt Family Member 10A), the likely causal gene at the chr.2q35 risk locus for which a functional interaction with another MPHL risk locus has been shown[31]. These findings suggest that both common and rare variation in these genes contributes to MPHL etiology. The analyses further identified an association with *HEPH* (Hephaestin), which, while being located less than 500 kb upstream of *EDA2R*, has not been previously considered a candidate gene. However, recent reports have indicated that *HEPH* plays a crucial role in hair development through its ferroxidase activity[32]. In addition to the insights that our rare coding variant analyses yielded at GWAS loci, they also implicate novel MPHL candidate genes beyond GWAS loci, namely *CEPT1* (Choline/ethanolamine phosphotransferase 1) and *EIF3F* (Eukaryotic translation initiation factor 3 subunit F). *CEPT1* encodes the terminal enzyme in the Kennedy pathway of phospholipid

biosynthesis[33]. While no reports specifically linking *CEPT1* and hair (loss) biology exist, there is evidence for a link between phospholipid metabolism and hair biology. For example, the topical administration of phospholipids was shown to promote hair growth in mice[34], and overexpression of group X-secreted phospholipase A$_2$ in mice led to alopecia and changes in hair cycling[35]. *EIF3F* encodes a subunit of the eukaryotic initiation factor 3 (eIF-3) complex. Recent reports suggest a potential involvement of *EIF3F* in hair pigmentation, as a patient with two heterozygous variants in *EIF3F* presented with skin and hair hypopigmentation[36], and a heterozygous *EIF3F* knock-out resulted in abnormal coat pigmentation in mice[37]. This is of interest as the transformation of pigmented terminal hair follicles to unpigmented vellus hair follicles is a pathophysiological feature of MPHL[38]. Additionally, *EIF3F* has been shown to act as a negative regulator of cell proliferation in cancer cells[39], and was shown to regulate Notch signaling[40], which in turn is involved in hair follicle stem cell fate determination[41].

Our conditional single-variant analysis further identified a number of strong associations independent from common GWAS variants. Among the top ten variant associations from this analysis are variants located within the genes *AR* (androgen receptor), *WNT10A*, *TSPEAR* (Thrombospondin Type Laminin G Domain and EAR Repeats), *MITF* (Melanocyte Inducing Transcription Factor) and *DGKZ* (Diacylglycerol Kinase Zeta). Given that these rare, nonsynonymous coding variants achieved low *P*-values - albeit above the threshold for genome-wide significance - despite the generally low power of the single-variant analyses, these may constitute independent candidate genes. The two genome-wide significant single variant associations 23:66604439:G:A (in *EDA2R*) and 23:66197712:C:T (in *HEPH*) did not retain genome-wide significance after conditioning, pointing to a (partial) interdependence between these variants and common GWAS variants, which was more pronounced for 23:66197712:C:T, while 23:66604439:G:A retained a partial signal. We further observed that (i) the rare MPHL risk allele of the 23:66604439:G:A variant occurs

**Table 1 | Results of conditional SKAT-O analysis, involving the removal of the two variants that showed genome-wide significance in the single-variant analyses (23:66197712:C:T and 23:66604439:G:A)**

|  | Gene | *P* | *P*$_{conditioned}$ |
|---|---|---|---|
| Continuous model | *HEPH* | $7.3 \times 10^{-9}$ | $1.3 \times 10^{-4}$ |
|  | *EDA2R* | $1.4 \times 10^{-8}$ | 0.84 |
| All-model | *HEPH* | $1.5 \times 10^{-7}$ | $1.3 \times 10^{-2}$ |
|  | *EDA2R* | $1.4 \times 10^{-7}$ | 0.79 |
| Two-as-control model | *HEPH* | $1.7 \times 10^{-7}$ | $2.8 \times 10^{-5}$ |
|  | *EDA2R* | $5.9 \times 10^{-6}$ | 0.81 |
| Extreme model | *HEPH* | $2.0 \times 10^{-7}$ | $5.7 \times 10^{-6}$ |
|  | *EDA2R* | $8.0 \times 10^{-5}$ | 0.12 |

The SKAT-O *P*-value (unadjusted) before and after conditioning is shown according to gene and phenotype model.

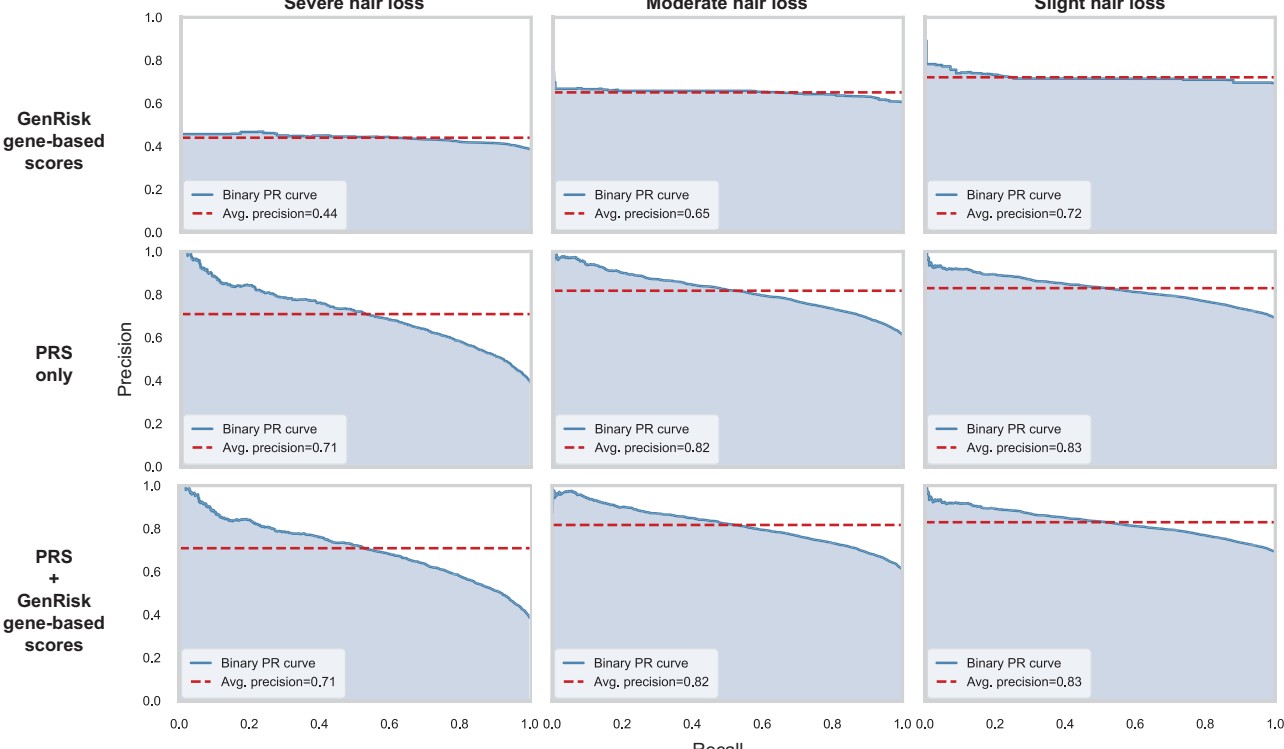

**Fig. 7 | Precision-recall-curves of the created MPHL risk models based on PRS only, GenRisk gene-based scores, and PRS combined with GenRisk gene-based scores.** The models were tested in terms of prediction of no hair loss (pattern 1) vs severe hair loss (pattern 4), at least moderate hair loss (pattern 3-4), and at least slight hair loss (pattern 2-4). PRS polygenic risk score, PR precision-recall, Avg. average.

exclusively on the common MPHL risk haplotype previously reported by Hillmer et al.[42] (rs2497935-A, rs962458-A, rs12007229-C, rs12396249-G) and (ii) the rare protective allele of the 23:66197712:C:T variant occurs almost exclusively on a lower-risk haplotype with only the rs962458-A risk allele. While the 23:66604439:G:A variant exclusively occurs on the previously reported MPHL risk haplotype, a partial signal remains in the conditional analyses, which may point to an independent effect of the rare variant and the risk haplotype. However, at this point, a causal role of either variant can neither be confirmed nor excluded.

As genes identified through our rare variant gene-based association tests were enriched for genes at known MPHL GWAS loci (lead SNP ± 1 Mb), these data underline the importance of studies that assess the entire allelic spectrum of disease associations, and their potential to highlight causal genes at GWAS risk loci. A conditional GWAS-GenRisk analysis was performed and found no systematic dependence between common GWAS-implicated variants and GenRisk gene scores. The analysis however identified risk loci where the GWAS association signal appears to be (partially) driven by both common and rare variants, namely chr.Xq12 (*EDA2R*) and chr.2q35 (*WNT10A*). Associated loci which are not impacted by any GenRisk gene score may be due to a low contribution of rare deleterious variants to the association. However, further investigation into the extent of dependence between common variants and GenRisk gene scores is required.

The X-chromosome has long been at the center of genetic analyses on MPHL. Early studies focused on the X-linked androgen receptor gene (*AR*), due to the strict androgen dependency of the phenotype. Although the results have been conflicting in regards to the likely causal variants and genes, the *AR*/*EDA2R* locus has consistently been the most strongly associated genomic region for MPHL, although neither the precise causal variants nor the causal genes have been confirmed[43]. In the present study, we identified significant associations with two X-chromosomal genes, namely *EDA2R and HEPH*, thereby yielding new or additional evidence for these candidate genes. Our analyses did not identify significant associations of rare variants in the *AR* gene ($P_{SKAT-O\ binary} = 7.6 \times 10^{-5}$). This is in line with previous Sanger-sequencing-based studies of the *AR* coding sequence, which did not identify any significant associations between the *AR* and MPHL[44,45]. Although we cannot exclude the possibility that our analysis lacked statistical power to detect such an association, one might also speculate that a potential involvement of the *AR* gene in MPHL pathobiology is impacted primarily by regulatory common variants, rather than rare variants in or around its coding sequence.

Moreover, a less stringent set of MPHL-associated genes overlapped with and were enriched for genes that have been reported as the cause of monogenic trichoses, namely *WNT10A* (odonto-onychodermal dysplasia and Schöpf-Schulz-Passarge syndrome), *HOXC13* (pure hair and nail ectodermal dysplasia), *DSP* (Carvajal syndrome), *LPAR6* (hypotrichosis 6), *ALX4* (total alopecia in frontonasal dysplasia), *EDAR* (ectodermal dysplasia), *CDH3* (ectodermal dysplasia), *HR* (hypotrichosis 4 and alopecia universalis), and *SPINK5* (Netherton syndrome). Notably, most of these genes either cause ectodermal dysplasias, hypotrichoses or alopecia. However, as we detected variants with a previously reported likely or known pathogenic association with genotrichoses in both cases and controls, no definitive statement can be made as to whether the presence of or variable expressivity of a genotrichosis may have led to a misclassification in the MPHL self-report. Generally, an overlap between genotrichoses- and MPHL-associated genes would be biologically plausible, as different levels of impairment of key hair follicle signaling pathways would be expected to result in differing phenotypes. For example, GWAS have previously yielded evidence for an association between hair curl and MPHL[9]. Together, these findings may indicate an overlap in causal genes between genotrichoses and MPHL.

Rare coding variants in the associated genes identified in this study have been previously associated with phenotypes such as mean corpuscular haemoglobin (*EIF3F*) and urea (*HEPH*)[46,47]. Suggestive associations have further been identified between testosterone levels and *EDA2R*, and alcohol use and *EIF3F*. Some of these associations may present interesting links – for instance, epidemiological studies have (albeit with conflicting evidence) found associations between MPHL and alcohol consumption[48].

The present analyses utilized four different phenotype models. Our continuous model represented a 1:1 representation of the progressive phenotype, which may however be most sensitive to misclassifications in the self-report. Our all-model provided a simple description of the phenotype by considering unaffected men as controls and men with any type of balding (frontal or vertex) as cases. The purpose of our extreme model was to achieve complete separation between cases and controls, despite the age-dependent and progressive nature of MPHL. This involved considering men with complete baldness of the scalp below 60 years of age as cases, and unaffected men aged 60 years or older as controls. The aim of this approach is to facilitate detection of variants and genes contributing to balding in relatively younger men and may provide higher statistical power, as these supercontrols are among the 10% of men least affected by MPHL[1] and are unlikely to develop a significant degree of balding during their lifetime. However, this phenotype model comes at the expense of sample size, which was reduced by nearly 80% compared to the other phenotype models. The purpose of the two-as-control model was to address the possibility of misclassifications in the self-reporting of balding. Misclassifications may be possible for UK Biobank MPHL patterns 1 and 2 (unaffected vs frontal balding), since we are of the opinion that the presence of balding in the frontotemporal regions of the scalp may be subjectively over- or underestimated in the absence of a dermatological assessment. In the present study, the different phenotype models yielded partially distinct gene associations, for example *WNT10A* and *EIF3F*, which consistently showed stronger signals in the two-as-control model. This may be an indication that distinct mechanisms contribute to more severe stages of balding, which are easier to detect using this case-control separation. All in all, the phenotype models employed in this study provide different perspectives on the MPHL phenotype and can account for certain possible errors in the self-report.

In this study, we performed two types of gene-based analyses: SKAT-O and GenRisk. SKAT-O is a well-established tool for gene-based association analyses and has the ability to detect associations in the presence of mixed effect directions at the variant level. GenRisk employs a scoring system that uses a beta distribution weighting schema for allele frequency, which is similar to SKAT-O, and pathogenicity scores (CADD score), to upweight rare and deleterious variants. As a result, GenRisk does not require variant consequence filtering. Moreover, GenRisk generates individual-level gene-based scores, which can be used in downstream analyses such as association analyses and risk prediction modeling. GenRisk was recently used to identify associations between rare genetic variants and blood biomarkers, identifying both known and novel associations (preprint)[49]. In the present study, both methods yielded partially distinct gene associations. While the inclusion of non-coding variants and non-protein-coding genes in the GenRisk analysis may yield overall more comprehensive results, the association signal may encompass a greater overlap with GWAS. The GenRisk analysis of coding variants only, on the other hand, offers an increased focus on high-impact coding variants, without severely reducing the number of variants through e.g., high-impact variant consequence filters. The analyses employed in this study therefore address different hypotheses. While each method offers different biological insights, some identified gene associations are consistent between SKAT-O

and GenRisk, and Fisher's exact tests show a significant overlap of a less stringent set of associations ($P < 3 \times 10^{-3}$) between the two analyses across all phenotype models ($OR_{continuous} = 54.3$, $P_{continuous} = 1.5 \times 10^{-17}$; $OR_{all\text{-}model} = 92.8$, $P_{all\text{-}model} = 2.1 \times 10^{-24}$; $OR_{two\text{-}as\text{-}control} = 71.3$, $P_{two\text{-}as\text{-}control} = 3.5 \times 10^{-20}$; $OR_{extreme\ model} = 99.1$, $P_{extreme\ model} = 3.9 \times 10^{-19}$). However, given the novelty of the approach, corroboration of the GenRisk results in further studies is desirable.

To our knowledge the present study represents the first systematic analysis of the contribution of rare variants to MPHL etiology. While rare variants in coding regions of the genome seem to make only a small contribution to MPHL genetic risk at population-level and may have little value for risk prediction, they may nonetheless contribute significantly to individual risk. In line with this hypothesis, we observed only a marginal contribution to the overall MPHL risk prediction of gene-based burden scores with respect to the PRS. Since prediction model performances are typically assessed on overall data set metrics (such as AUC) it can be expected that the impact of variables informative only for a small proportion of samples can be marginal (e.g., there might be few individuals whose MPHL genetic risk can be attributed to damaging rare variants in a specific MPHL susceptibility gene). Instead, PRS by providing a gradient-risk in the overall data set can model the genetic risk throughout the population, therefore representing a global genetic risk variable. While gene-based burden scores may not be particularly suited for risk prediction models in the general population, they are a powerful instrument to detect gene associations and can therefore be helpful to dissect the genetic architecture of complex traits such as MPHL. As demonstrated with our study, the analysis of rare variants additionally offers important insights into associated alleles, genes and pathways, as well as pleiotropy, thereby improving our understanding of MPHL pathobiology. While the present study provides first insights into the contribution of rare variants to MPHL pathobiology based on a tranche of 200,629 exomes from the UK Biobank, the final data set of ~450,000 exomes has been released while completing the present analyses. This data set represents a considerable increase in sample size. Continued investigation on the role of rare variants for MPHL using this larger data set is therefore warranted.

In summary, the findings of our analysis broaden the allelic spectrum of previously reported candidate genes (*EDA2R*, *WNT10A*), yield evidence for novel MPHL candidate genes both at (*HEPH*) and beyond (*CEPT1*, *EIF3F*) known GWAS loci and suggest an association between genotrichoses and the common MPHL phenotype. Together, they provide a basis for future investigations into MPHL pathobiology and the contribution of rare variants to MPHL. Investigations of the functional relevance of rare variants and their interactions with common variants at and beyond risk loci will eventually improve our understanding of MPHL pathobiology and may lead to improved risk prediction and identification of affected pathways and can pave the way for the development of personalized therapies.

## Methods

### Phenotype data

The UK Biobank study has been approved by the North West Multicentre Research Ethics Committee as a Research Tissue Bank and all UKB participants provided written informed consent. The UKB 200k release contains exome- and MPHL self-report data from 89,311 men[18,19]. These MPHL self-report data were recorded at up to four UKB assessment center visits. Using a touch-screen questionnaire, participants scored their hair loss on a scale of 1 to 4, as based on four pictograms (Supplementary Fig. 8): 1 – Unaffected; 2 – frontotemporal balding; 3 – balding of the frontotemporal region and vertex; and 4 – complete baldness of the top of the scalp.

In the present study, four phenotype models were used: (i) a continuous model, which considers hair loss patterns 1–4 on a

continuous scale, (ii) an all-model, in which controls (pattern 1) were compared to cases (pattern 2–4); (iii) an extreme model, in which supercontrols (pattern 1, age ≥60) were compared to severe cases (pattern 4, age <60); and (iv) a two-as-control model, in which controls (pattern 1–2) were compared to cases (pattern 3–4) in order to address the possibility of misclassifications between pattern 1 and 2 in the self-assessment.

For individuals who provided MPHL data at more than one assessment center visit, additional steps were performed in order to check the self-report data for sanity, and to select an entry for use in the analyses. The most recently recorded MPHL pattern was selected for analysis, unless an improvement in MPHL status was recorded. Due to the progressive nature of MPHL, an improvement is implausible. To avoid the need to exclude individuals who reported an improved MPHL status and to instead identify a plausible MPHL pattern, the following steps were performed: (i) if two balding patterns were available, and the difference between the patterns was no larger than 1, the higher pattern was used; (ii) if 3 balding patterns were available, a pattern that was recorded 2 times was used; (iii) if 4 balding patterns were available, a pattern that was recorded 3 times was used. If no plausible MPHL pattern could be identified in this manner, the individual was excluded. To account for the age-dependency of MPHL, in case of multiple assessments, for cases, we selected the lowest age at which the highest MPHL pattern was recorded. For controls, we selected the highest age at which no (pattern 1) or mild (pattern 2) hair loss was recorded.

To select participants for the present analysis, the following four criteria were used: (i) no grounds for exclusion found in the MPHL multi-entry sanity check; (ii) availability of genotype and kinship data; (iii) genetically and self-reported male sex with no sex chromosome aneuploidy; and (iv) self-reported white British ethnicity, as well as very similar genetic ancestry, as based on a principal components analysis of the genotypes. In addition, related individuals up to the third degree were excluded on the basis of UKB kinship coefficients (kinship coefficient ≥0.0442). Iterative exclusion was performed for one individual in a related pair, with individuals with a larger number of related individuals being excluded preferentially. An unexpected improvement of MPHL was observed in 2235 individuals. Of these, 293 were excluded since no plausible MPHL pattern could be nominated. A total of 72,469 of the 89,311 male UKB participants fulfilled these criteria. Exclusion of related individuals was performed separately for each phenotype model, resulting in the following final sample counts: 72,024 (continuous, all- and two-as-control models) and 17,053 (extreme model).

### Variant data

Exome sequencing variant data for the 200,643 participants in the UKB 200k release were downloaded from the UKB in PLINK format. The data comprised 17,981,897 variants, which were captured from 204,829 autosomal and gonosomal exonic regions ±100 bp flanking regions. For the SKAT-O and single-variant analyses, the data were quality controlled in PLINK 2.0[50] with respect to per-individual missing rate (<5%), per-variant missing rate (<5%), and Hardy-Weinberg equilibrium ($P > 10^{-6}$). Variants were filtered for a MAF < 1% based on their frequency in the different phenotype model data subsets. Variants were converted to variant call format (VCF) and annotated using the Ensembl Variant Effect Predictor (VEP)(v104)[51]. Variants with a predicted nonsynonymous consequence of moderate or high impact in a protein-coding gene (as based on Ensembl gene annotation release 104) were selected. These variant criteria comprised missense, insertion, deletion, splice acceptor-, splice donor-, and start- or stop-altering variants, as well as transcript and regulatory region ablations.

For the GenRisk analyses, variant data were quality controlled in PLINK 2.0 with respect to per-variant missing rate (<2%) and

Hardy-Weinberg equilibrium ($P > 10^{-6}$). Variants were filtered for an MAF < 1%, as based on their frequency in the data set. The variant data were annotated with CADD (Combined Annotation Dependent Depletion) scores (CADD v1.6)[52] and gene features based on the GRCh38 NCBI RefSeq refFlat table from the UCSC Genome Browser[53,54].

Imputed genotype data were downloaded from the UKB in BGEN format. These data comprised information concerning 97,059,328 variants and were converted to PLINK format using PLINK 2.0 and the *ref-first* parameter.

## Correction for population stratification

To account for population stratification, analyses were performed to estimate the optimal number of top PCs to include in our statistical models. For this purpose, GWAS were performed on the UKB imputed genotype data using a varying number of included PCs, and the genomic inflation factor λ was determined. Imputed genotype data were processed in PLINK 2.0, with preservation of the imputed genotype dosages. The data were quality controlled for info score (≥0.3) and minor allele count (≥20) and filtered for each of the four phenotype model subsets. GWAS were performed in PLINK 2.0, with correction for age and the 1–20 top PCs, as pre-calculated by the UKB based on imputed SNP genotype data. In the extreme model, age correction was omitted, since this phenotype model differentiates based on age.

## Association analyses

The association analyses of the continuous model, all-model and the two-as-control model were corrected for age, sequencing batch and the top 14 PCs. In the association analyses of the extreme model, correction was made for sequencing batch and the top five PCs only. GWAS-style single variant analyses of the filtered exome data were performed in PLINK 2.0 using the *glm* function with covariate normalization to mean 0, variance 1. LD of the single-variant associations with the sentinel GWAS SNP was calculated in PLINK 2.0 using the *ld* function. For the 23:66197712:C:T variant in *HEPH*, the nearest GWAS lead SNP was used: 23:66001818:T:A (rs771150309, MPHL risk allele = major allele = T). For the 23:66604439:G:A variant in *EDA2R*, the closest GWAS lead SNP (23:66418642 (rs5965195)) was not contained in the employed imputed genotyping data release, and the nearest significant SNP was used instead: 23:66418267:G:A (rs4827473, MPHL risk allele = major allele = A).

Two types of gene-based analyses were performed: SKAT-O[20] and GenRisk[21]. SKAT-O was applied to the filtered exome data using the *SKATBinary.SSD.All* (for binary phenotype definitions) and the *SKAT.SSD.All* (for the continuous phenotype definition) functions with default settings in the SKAT R package (v2.0.1)[20]. Data were converted to PLINK 1 binary format using PLINK 2.0 for use as input files. Variants were assigned to genes based on the VEP annotation approach described above. In addition to the nonsynonymous variant consequence threshold imposed through the present filtering steps, more stringent thresholds were applied in this analysis by restricting inclusion to variants of high impact, as based on VEP annotation (splice acceptor, splice donor, stop- or start-altering and frameshift variants, as well as transcript ablations). The GenRisk analysis was performed on the filtered exome data in VCF format using the GenRisk Python package (v0.2.5)[21]. GenRisk was applied separately to i) rare variants (MAF < 1%) annotated to any gene and ii) only coding variants, using the identical variant set as used in the SKAT-O analyses. Gene-based scores were generated using weighted MAF (beta density function with parameters a = 1 and b = 25) and raw CADD scores as functional annotation, whereby variants with a lower MAF or higher CADD score were upweighted. The association analysis of the gene-based scores and previously described covariates was performed using linear regression (continuous model) or L1-logistic regression (all-, two-as-control and extreme models).

The *P*-value threshold for genome-wide significance in single-variant association analyses was selected as $8 \times 10^{-9}$, as empirically determined by Karczewski et al. based on analyses of 394,841 UK Biobank exomes[46]. *P*-value thresholds for the SKAT-O and GenRisk gene-based analyses were determined using Bonferroni correction based on the maximum number of genes tested, resulting in a threshold of $2.6 \times 10^{-6}$ (corresponding to 18,946 genes tested in the SKAT-O analysis).

## Enrichment analyses

To improve the feasibility of enrichment analyses and obtain a more comprehensive gene list of approximately 500 genes, a less stringent *P*-value threshold of $P < 3 \times 10^{-3}$ was selected, resulting in a less stringent set of 595 MPHL-associated genes. Testing was performed for an enrichment of this less stringent set of MPHL-associated genes in genes located ±1 Mb of previously published GWAS lead SNPs[7–17]. Enrichment testing using a one-tailed Fisher's exact test from scipy (v1.8.1) was performed with a background list comprising the final tested genes per phenotype model. Using the same method, analyses were also performed to test for an enrichment of this less stringent set of MPHL-associated genes in genes causative for monogenic trichoses. A list of 65 known trichosis genes was created, as based on previous publications[23–27]. The genes and their corresponding condition are listed in Supplementary Table 1.

## ClinVar query

An inspection was made to determine whether MPHL-associated rare variants have been described as pathogenic or likely pathogenic on ClinVar. ClinVar data were downloaded as VCF (accessed 02.05.2022) and filtered for nominally significant single variants ($P < 0.05$ in any phenotype model). Information on associated conditions was extracted for variants listing a clinical significance of *pathogenic*, *likely pathogenic* or *conflicting interpretations of pathogenicity*.

## Conditional analyses

To evaluate the dependence of association signals on specific variants, conditional analyses were performed. Gene-level conditional analyses of the genes *EDA2R* and *HEPH* were conducted using SKAT-O, as previously described, after the exclusion of two variants that showed genome-wide significance in the single-variant analyses (23:66197712:C:T and 23:66604439:G:A).

Using data from the continuous model, we tested whether SNPs previously implicated in GWAS were independent from GenRisk gene scores. Imputed genotype data from the UKB were filtered for MAF (>1%), info score (>0.3), per-variant missing rate (<5%), Hardy-Weinberg equilibrium ($P > 10^{-6}$). Association analyses of the filtered imputed genotype data were performed in PLINK 2.0 using the *glm* function with covariate normalization to mean 0, variance 1, and corrected for age and 14 top PCs. The analyses were performed per locus, defined based on 622 SNPs that were identified as independent MPHL lead SNPs in a UKB-based GWAS[13] ± 500 kb flanking regions. For each gene per locus, the analysis was additionally corrected for the respective GenRisk gene scores. Resulting *P*-values and effect sizes were then compared between the uncorrected and gene-corrected analyses.

To test whether the rare single-variant associations were independent from common SNPs previously implicated in GWAS, imputed genotype data from the UKB were filtered for info score (>0.3) and 622 SNPs that were identified as independent MPHL lead SNPs in a UKB-based GWAS[13]. GWAS-style exome single variant analyses were then performed as described above, with the inclusion of genotypes for lead SNPs on the same chromosome as covariates.

## Pathway gene set and network analyses

Gene set analysis was performed using FUMA GENE2FUNC[55] (v1.4.0) with default settings. MPHL-associated genes ($P < 3 \times 10^{-3}$ in any gene-

based test) were used as input gene list. All tested genes were supplied as a background list. The results were filtered for pathway gene set categories, namely canonical pathways, curated gene sets, computational gene sets, chemical and genetic perturbation, hallmark gene sets, Reactome, KEGG and WikiPathways. To further obtain an overview of protein interactions and co-expression, a STRING (v11.5)[56] protein network analysis was performed using a less stringent set of MPHL-associated genes. Here, a threshold of $P < 3 \times 10^{-4}$ was selected in order to obtain a more manageable network of <100 genes.

### Risk modeling

To test whether the inclusion of rare variants improves common variant-based risk modeling of MPHL, GenRisk was used to create a risk prediction model integrating MPHL PRS with GenRisk gene-based scores, which were generated as described above. In order to establish a PRS model, a GWAS of imputed genotyping data from the UKB was performed based on data from our continuous model. Individuals with no exome sequencing data were selected to ensure no sample overlap, and filtered using the criteria described previously, resulting in 105,565 unrelated (both within this sample and with the 72,024 individuals of the continuous, all- and two-as-control models) male individuals. The imputed genotype data were quality-controlled (info score >0.3, per-variant missing rate <5%, HWE $P > 10^{-6}$) and filtered for common variants (MAF > 1%). The GWAS was performed in PLINK 2.0 using the *glm* function, and corrected for age and 18 top PCs (estimated as the optimal number of top PCs for this sample based on λ calculation).

PRS were calculated for the cohort of 72,024 males using PRSice-2 (v2.3.5)[57] using autosomal and X-chromosomal SNPs $P < 7.85 \times 10^{-3}$ (best-fit PRS *P*-value threshold) and otherwise default settings. AUCs for the full cohort were computed using the pROC R package (v1.18)[58]. The cohort of 72,024 males was split 25–50–25%, with 25% being used for weighting genes and summing all gene-based scores into one gene-based risk score per individual. Training of the integrated risk prediction models was performed using 50% of the samples with 10-fold cross-validation, with the remaining 25% of samples being used as an independent testing set. The risk prediction model was generated based on data from our continuous model with age, sequencing batch and the top 14 PCs being included as features. To evaluate the contribution of rare variants, the performances of risk models that included gene-based scores and PRS were compared with risk models that included PRS only.

### Reporting summary

Further information on research design is available in the Nature Portfolio Reporting Summary linked to this article.

## Data availability

This research has been conducted using data from UK Biobank under Application Numbers 24661 and 102444. The individual-level genetic and phenotypic data are available under restricted access; access can be obtained by application through the UK Biobank platform. The data generated that support the findings of this study are provided in the Supplementary Data. The CADD score data used in this study are available in the University of Washington CADD score database https://krishna.gs.washington.edu/download/CADD/v1.6/GRCh38/whole_genome_SNVs.tsv.gz. The gene feature annotation data used in this study are available in the Ensembl database under release number 104 https://ftp.ensembl.org/pub/release-104/gtf/homo_sapiens/Homo_sapiens.GRCh38.104.chr.gtf.gz and in the UCSC Genome Browser https://hgdownload.soe.ucsc.edu/goldenPath/hg38/database/refFlat.txt.gz. The ClinVar data used in this study are available from the ClinVar database https://ftp.ncbi.nlm.nih.gov/pub/clinvar/vcf_GRCh38/archive_2.0/2022/clinvar_20220430.vcf.gz.

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

## Acknowledgements

This research was conducted using the UK Biobank resource under Application Numbers 24661 and 102444. We would like to thank the Core Unit for Bioinformatics Data Analysis for providing computing resources and support in setting up the initial pipeline. We further thank Christine Schmäl for her proofreading and feedback on the manuscript.

## Author contributions

L.M.H., O.B., P.M.K., C.M., M.M.N., S.H.-H. conceived and designed the research goals and analyses. S.H.-H. supervised the project. P.M.K. provided computing resources and analysis tools. S.K.H., S.S., R.A. curated the data. S.K.H., R.A., S.S., O.B., C.M. performed the analyses. S.K.H. wrote the initial manuscript draft and S.K.H., R.A., L.M.H., C.M., M.M.N., S.H.-H. revised the manuscript.

## Funding

## Competing interests

The authors declare the following competing interests: M.M.N., S.H.-H. and L.M.H. receive salary payments from Life & Brain GmbH and M.M.N. holds shares in Life & Brain GmbH. M.M.N. has received fees for membership in an Advisory Board of HMG Systems Engineering GmbH, and for membership in the Medical-Scientific Editorial Office of the Deutsches Ärzteblatt. M.M.N. is a member of the excellence cluster ImmunoSensation[2]. All this concerned activities outside the submitted work. S.K.H., R.A., S.S., O.B., P.M.K. and C.M. declare no competing interests.
