## [Peer Review File · Nature Communications]

REVIEWER COMMENTS

Reviewer #1 (Remarks to the Author):

This manuscript presents an extensive analysis into the contribution of rare (MAF<1%) variants to male pattern hair loss (MPHL) using a data set of 72k men from the UK Biobank with exome sequencing and (self-reported) MPHL phenotype data available. While several GWAS of MPHL have been conducted, the largest ones based on the UK Biobank array genotype data, these have primarily captured the common variant signal, therefore motivating such an exploration into the rare end of the genetic variation spectrum.

The analyses of individual variants pinpoint only two significant rare variants associated with MPHL, both previously identified in GWAS, but gene-based tests appear to point also to potential new candidate genes. All in all, results from the rare variant analyses suggest a minimal role of rare variants in MPHL heritability, aligned with predictions based on imputed genotype data. Therefore, although unique in its setting, the manuscript presents a limited number of novel insights into MPHL.

While the manuscript includes comprehensive analyses into the rare variant burden in MPHL, I have nevertheless identified several concerns and unclarities, primarily concerning statistical tests and rigour, and therefore provide below quite a few comments and suggestions to the authors on how to improve the work:

How was the significance threshold ($P < 5e-8$) for single variant analyses derived? Is the traditional genome-wide significance threshold appropriate for rare variant studies? How was the significance threshold for gene-based tests selected?

Given that the authors have predefined a significance threshold for their analyses, what is the justification for reporting nominally significant or suggestive associations in the manuscript main text?

How many variants were included in the single variant analyses and how were these selected? E.g. I gather singletons were excluded. However, this is not specified and I have trouble linking the numbers in Table 1 to the numbers reported in the manuscript (lines 96-101). Consider adding numbers of included variants and genes for all three analyses in Figure 1. I also recommend moving Table 1 to the supplementary material, as currently this seems to provide details that are irrelevant to the key story of the manuscript. E.g., the information in the "N ultra-rare variants" or "CADD range" columns are not to described in the main text at all.

Report variant consequences, allele frequencies and effect sizes for the two significant single variant associations in the main text. Additionally, report the LD of these variants to the GWAS sentinel SNPs in their respective loci.

Can one draw a conclusion from these results that these individual rare variant associations are irrelevant (which is the impression I was left with after reading the manuscript)? Could these represent the causal variants underlying the GWAS peaks? I recommend elaborating and clarifying the discussion related to the connection between GWAS and rare variant associations in these loci.

Based on Figure 2a, it seems the MPHL pattern groups differ in their mean ages (Density plot might represent the underlying age dependency better than the boxplot currently used). How were the age differences between the groups taken into account? The methods description of this is brief and I am unsure if such an adjustment is sufficient. The largest GWAS of MPHL published, also based on the UK Biobank (Yap, Nat Commun, 2018) used the pattern groups as score (taking values 1-4) that was adjusted for age, PCs etc. Did the authors consider using a similar continuous variable to boost their power?

Since three different MPHL phenotypes are used in the analyses, could comparisons between the rare variant associations and/or gene-based results between the three phenotype definitions allow to draw inferences on the correct phenotypic architecture of MPB (e.g., a continuous phenotype or distinct phenotypes)?

The authors applied two gene-based tests in the manuscript, SKAT-O, which is a well-established test (like the authors note in the discussion), and a newer method called GenRisk. As the GenRisk test is not widely used in the community, the key idea of the test and how it differs from SKAT-O should be described in the result section (Gene-based association analyses).

I find it concerning that the results from SKAT-O and GenRisk differ so substantially and therefore consider the downstream analyses based on GenRisk potentially dubious given the large number of gene associations found. Is the 7% overlap significant or rather expected by chance? Could the GenRisk analysis be conducted with the identical set of variants as SKAT-O to better allow for comparisons between the two approaches and to alleviate the concerns regarding the use of GenRisk?

Which data set the PRS data was from and which data set was used to test the predictive performance of PRS and GenRisk scores? Can the authors be sure there is no bias from overlapping samples?

Previous work has shown an enrichment of X-chromosomal associations for MPB and the two lead associations discovered in this work also reside in the X chromosome. The authors should include discussion related to the relevance of the X chromosome in MBP in the light of their findings.

The lack of any associations for AR is fairly surprising, e.g., given that AR missense burden shows an association with testosterone (genebase.org). How many variants were included in the burden tests for AR? How large an effect would have been detected in the 72k men, e.g., with 80% power?

The second paragraph of the discussion section is overly long and hence exhaustive to follow. Consider cutting text or splitting it into several paragraphs.

Line 164 mentions expression in hair follicles. Which threshold was used to determine expression in this tissue type? Are the genes specific in their expression in hair follicles? How many genes expressed in hair follicles would one expect by chance, i.e., is there any enrichment among the selected genes?

Supplementary tables for gene-based tests identify the genes using gene symbols only. I recommend adding, e.g., ENSEMBL IDs, to avoid potential Excel conversion issues.

Reviewer #2 (Remarks to the Author):

This manuscript explores the contribution of rare (minor allele frequency $\leq 1\%$) variation based on the subset of 72,469 exomes from the UK Biobank ~200k exome release to male pattern hair loss (MPHL). The authors perform single variant and gene-based analyses for three hair loss models identifying 124 significantly associated genes at FDR < 0.05. These genes are then taken forward for a number of follow-up analyses, although there is some concern that the majority of the 124 genes are only detected using one analysis method.

1. The current results are based on 200k exomes yet UK Biobank have made the full release of 450k exomes available through the Research Analysis Platform. These have been available for a while and would substantially increase the sample size for these analyses. Have the authors looked at that data?

2. Gene-based analyses were performed using two methods, SKAT-O and GenRisk. I note that one of the authors has developed GenRisk, but it is not entirely clear why both methods were used, what the differences are, and how they lead to such different results. Given the sample sizes, the results from

GenRisk seem to be inflated, did the authors look at inflation estimates (λ) for both methods? SKAT is a well-established method for gene-based analyses and it is possible to include weights in the analysis, did the authors compare results using a comparable weighting scheme in SKAT? Other than the weights, were the variants included in both analyses the same? I think it would be very important to thoroughly quality-control the results as the majority of follow-up analyses are based on the 124 significantly associated genes ($FDR < 0.05$) from the GenRisk results.

3. For example, the top association for GenRisk is YIPF6, did the authors look at what is driving that association? Interestingly it is not mentioned in the manuscript.

4. Does the Conditional GWAS-GenRisk analysis suggest that the GenRisk signal for genes where the GenRisk score attenuates the GWAS signal, are being driven by the GWAS variants? (or variants in linkage disequilibrium).

5. The variants identified in the single variant analyses and included in the conditional analyses (Table 2) have $D' = 1$ (in LDlink), which is consistent with a haplotype effect reported in Hillmer et al., 2009: <https://doi.org/10.1007/s00439-009-0668-z>. This would make the conditional analyses much harder to interpret.

6. Did the authors check which other phenotypes highlighted genes were associated with in the public portals such as azpewas.com or genebass.org?

7. Did analyses include a covariate for sequencing batch? (50k vs. 150k)

8. Authors should also add a reference for the exomes, Szustakowski et al, (2021): <https://doi.org/10.1038/s41588-021-00885-0>

9. Figure 1. HWD is should be HWE (which is also consistent with the text, lines 91/92).

10. It would be helpful to include the rsids for variants in addition to the chromosome:position:alleles names.

11. Typo, line 226: Figure 99

Reviewer #1:

How was the significance threshold ($P < 5e-8$) for single variant analyses derived? Is the traditional genome-wide significance threshold appropriate for rare variant studies? How was the significance threshold for gene-based tests selected?

We thank the reviewer for these questions, which have prompted us to reconsider our significance thresholds. In the revised version of the manuscript, we have opted to use Bonferroni-correction for the gene-based tests (2.7×10^{-6} , corresponding to 18,848 genes tested in the GenRisk analysis, which is close to the 18,810 genes tested in the SKAT-O analysis) and the significance threshold of 8×10^{-9} suggested by Karczewski et al. (2022) for single-variant analyses. This threshold was derived from association analyses of 314 random heritable phenotypes based on ~400,000 UK Biobank exomes. While this may be more stringent than required for our study, it represents a reliable and conservative estimate of the multiple testing burden for single-variant analyses of the UK Biobank exome sequencing data. We have replaced the mention of FDR-correction in the Methods section with the following:

“The P-value threshold for genome-wide significance in single-variant association analyses was selected as 8×10^{-9} , as empirically determined by Karczewski et al. based on analyses of 394,841 UK Biobank exomes⁴¹. P-value thresholds for the SKAT-O and GenRisk gene-based analyses were determined using Bonferroni correction based on the maximum number of genes tested, resulting in a threshold of 2.7×10^{-6} (corresponding to 18,848 genes tested in the GenRisk analysis).”

Given that the authors have predefined a significance threshold for their analyses, what is the justification for reporting nominally significant or suggestive associations in the manuscript main text?

We thank the reviewer for this comment. We had originally chosen to report these associations, as experience has shown that less stringent thresholds may (with a note of caution) capture additional biologically plausible associations that the analyses were likely underpowered to detect. We have opted to keep using a less stringent set of associations for enrichment testing of gene sets, GWAS loci and genotrichosis genes as well as the ClinVar query, but have removed mentions of the number of nominally significant or suggestive associations in the manuscript main text.

How many variants were included in the single variant analyses and how were these selected? E.g. I gather singletons were excluded. However, this is not specified and I have trouble linking the numbers in Table 1 to the numbers reported in the manuscript (lines 96-101). Consider adding numbers of included variants and genes for all three analyses in Figure 1. I also recommend moving Table 1 to the supplementary material, as currently this seems to provide details that are irrelevant to the key story of the manuscript. E.g., the information in the "N ultra-rare variants" or "CADD range" columns are not described in the main text at all.

Variants for the single-variant analysis were selected identically to the variants for the SKAT-O analysis: Variants with a minor allele frequency $< 1\%$ and a nonsynonymous variant consequence in protein-coding genes were included in the analyses. Singletons were not excluded, and the mention of singletons in Table 1 was only intended as an overview of allele frequencies across the variant consequence categories. Numbers were understandably hard to link, as in many cases, one variant was annotated into several different consequence categories. As several columns were indeed not relevant to the key story of the manuscript, we have decided to remove Table 1 and instead included the numbers of variants and genes in Figure 1.

Report variant consequences, allele frequencies and effect sizes for the two significant single variant associations in the main text. Additionally, report the LD of these variants to the GWAS sentinel SNPs in their respective loci.

We thank the reviewer for this suggestion. We have added the information in the revised version of the manuscript. The relevant section now reads:

“The two genome-wide significant variants, i.e., 23:66604439:G:A (rs12837393, $MAF = 5.5 \times 10^{-3}$, $P_{\text{continuous}} = 3.0 \times 10^{-12}$, $\beta_{\text{continuous}} = 0.19$, $P_{\text{all}} = 4.8 \times 10^{-10}$, $\text{odds-ratio}_{\text{all}} = 1.53$, $r^2_{\text{sentinel SNP}} = 1.6 \times 10^{-4}$, $D'_{\text{sentinel SNP}} = 0.35$) and 23:66197712:C:T (rs151003259, $MAF = 2.0 \times 10^{-3}$, $P_{\text{continuous}} = 1.0 \times 10^{-13}$, $\beta_{\text{continuous}} = -0.35$, $P_{\text{all}} = 2.9 \times 10^{-10}$, $\text{odds-ratio}_{\text{all}} = 0.59$, $r^2_{\text{sentinel SNP}} = 4.5 \times 10^{-7}$, $D'_{\text{sentinel SNP}} = 1.0$)(GRCh38), are missense variants located within *EDA2R* and *HEPH* respectively. Notably, the T allele of 23:66197712:C:T was exclusively observed in combination with the MPHL risk allele ($MAF > 0.99$) of the respective GWAS sentinel SNP.”

Can one draw a conclusion from these results that these individual rare variant associations are irrelevant (which is the impression I was left with after reading the manuscript)? Could these represent the causal variants underlying the GWAS peaks? I recommend elaborating and clarifying the discussion related to the connection between GWAS and rare variant associations in these loci.

We thank the reviewer for this comment. Unfortunately, at this point, we can neither confirm nor exclude a causal role of the two individual rare variant associations in *EDA2R* and *HEPH*. Our conditional analyses point to an inter-dependence of these variants with the GWAS signal at these loci, although we cannot infer the directionality of this dependence.

We have experimented with fine-mapping (using the susieR tool) of a mixed data set of imputed genotyping data and exome sequencing data, and both rare variant associations yielded posterior inclusion probabilities of 0 even if the a-priori assumption of the causal number of SNPs was set relatively high (10-70). Notably however, the fusion of imputed genotyping data and exome sequencing data, as well as the analysis of X-chromosomal variants, are not common use cases.

We have added a paragraph to the discussion in which we aimed to clarify the connection between GWAS and rare variant associations at these loci (in the context of the haplotype constellations in which these rare variants occur): “The two genome-wide significant single variant associations 23:66604439:G:A (in *EDA2R*) and 23:66197712:C:T (in *HEPH*) did not retain genome-wide significance after conditioning, pointing to a (partial) inter-dependence between these variants and common GWAS variants, which was more pronounced for 23:66197712:C:T, while 23:66604439:G:A retained a partial signal. We further observed that i) the rare MPHL risk allele of the 23:66604439:G:A variant occurs exclusively on the common MPHL risk haplotype previously reported by Hillmer et al.³⁸ (rs2497935-A, rs962458-A, rs12007229-C, rs12396249-G) and ii) the rare protective allele of the 23:66197712:C:T variant occurs almost exclusively on a lower-risk haplotype with only the rs962458-A risk allele. While the 23:66604439:G:A variant exclusively occurs on the previously reported MPHL risk haplotype, a partial signal remains in the conditional analyses, which may point to an independent effect of the rare variant and the risk haplotype. However, at this point, a causal role of either variant can neither be confirmed nor excluded.”

Based on Figure 2a, it seems the MPHL pattern groups differ in their mean ages (Density plot might represent the underlying age dependency better than the boxplot currently used). How were the age differences between the groups taken into account? The methods description of this is brief and I am unsure if such an adjustment is sufficient. The largest GWAS of MPHL published, also based on the UK Biobank (Yap, Nat Commun, 2018) used the pattern groups as score (taking values 1-4) that was adjusted for age, PCs etc. Did the authors consider using a similar continuous variable to boost their power?

We thank the reviewer for these comments. We have adjusted Figure 2 to include a density plot of the age distributions per MPHL pattern group. Age was included as a covariate in all association analyses, except those performed for the ‘extreme’ phenotype model. This correction is in line with the most recent UK Biobank-based GWAS on MPHL employing a case-control phenotype classification (Pirastu et al., *Nat Commun*, 2017), and e.g., an adjustment for age² has been shown to not significantly improve (continuous) MPHL models (Yap et al., *Nat Commun*, 2019). We have further included a continuous phenotype model incorporating pattern values 1-4 in the revised version of the manuscript, and thank the reviewer for this suggestion.

Since three different MPHL phenotypes are used in the analyses, could comparisons between the rare variant associations and/or gene-based results between the three phenotype definitions allow to draw inferences on the correct phenotypic architecture of MPB (e.g., a continuous phenotype or distinct phenotypes)?

We thank the reviewer for this comment. The correct phenotypic architecture of MPHL has been a matter of some debate (e.g., whether early-onset MPHL is in fact distinct from late-onset MPHL), and it would be very interesting to differentiate whether genetic factors contribute to distinct mechanisms and parts of the phenotype, or whether they e.g., accelerate the rate of hair loss. Certain genes show consistently slightly stronger signal in certain phenotype models in our analyses (e.g. *WNT10A* and *RUNX3* in the two-as-control model vs the all-model), which may indicate that these genes contribute to relatively more severe degrees of balding. Although this is an interesting aspect, to us it appears that disentangling the phenotypic architecture of MPHL cannot be addressed within the UK Biobank cohort at the moment, but would instead require knowledge of the phenotypic endpoint, which is only feasible in retrospective studies.

The authors applied two gene-based tests in the manuscript, SKAT-O, which is a well-established test (like the authors note in the discussion), and a newer method called GenRisk. As the GenRisk test is not widely used in the community, the key idea of the test and how it differs from SKAT-O should be described in the result section (Gene-based association analyses).

We thank the reviewer for this suggestion. GenRisk is a package that generates gene-based burden scores for individuals. The scoring system uses beta distribution weighting schema for allele frequency, which is similar to SKAT-O, and pathogenicity scores (functional annotations), to upweight rare and deleterious variants. The pipeline creates gene-based scores on an individual level which can be used for further downstream analyses like association analysis and risk prediction modeling.

We agree that given the novelty of the approach, it would be better for the reader to have a more comprehensive description in the manuscript. We have expanded the section discussing the differences between both methods in the Discussion section, as well as adding a short description of the GenRisk tool in the results section:

“To assess the cumulative contribution of rare coding variants to MPHL, we performed gene-based association analyses using SKAT-O²¹ and GenRisk²², a new burden association test which upweights rarer and more deleterious variants (based on CADD), and can be particularly powerful in detecting deleterious gene associations.”

I find it concerning that the results from SKAT-O and GenRisk differ so substantially and therefore consider the downstream analyses based on GenRisk potentially dubious given the large number of gene associations found. Is the 7% overlap significant or rather expected by chance? Could the GenRisk analysis be conducted with the identical set of variants as SKAT-O to better allow for comparisons between the two approaches and to alleviate the concerns regarding the use of GenRisk?

We thank the reviewer for this comment. We further investigated this issue by comparing the nominally significantly associated genes ($P < 0.05$) between GenRisk and SKAT-O. At this p-value threshold, we observe a strong enrichment of shared associated genes between the 2 approaches (see Fisher's exact test results in the table below).

	Odds ratio	P-value
Continuous model	2.5	7.7e-24
All-model	2.9	1.2e-27

Two-as-control model	2.7	4.7e-24
Extreme model	2.4	2.0e-16

The originally reported 7% overlap is likely this low due to (i) very few associations overcoming the multiple comparison correction in the SKAT-O analyses and (ii) a number of genes previously being excluded from the GenRisk analyses (if the respective GenRisk gene scores were zero in a majority of the cohort). When removing the filter for zero rate and considering nominally significantly associated genes, there is a consistent overlap, though still with some heterogeneity of associations. In addition, by looking at the association with respect to genomic coordinates, there is a colocalization of the signal, since both GenRisk and SKAT-O detect strongly associated genes on the X-chromosome (in proximity to *AR*).

We agree that the reported 7% results could be misleading, as it might suggest that GenRisk and SKAT-O provide largely different results, thus raising doubts about the validity of secondary analyses. We have therefore removed the sentence with the reported 7% overlap and instead added a paragraph reporting the Fisher's test enrichment, while highlighting that there is heterogeneity in the association signal possibly due to the different methods implemented in the two tools (a polygenic burden test based on CADD and AF for GenRisk and a combined variance and burden test based on AF for SKAT-O).

Which data set the PRS data was from and which data set was used to test the predictive performance of PRS and GenRisk scores? Can the authors be sure there is no bias from overlapping samples?

We thank the reviewer for this comment. We have opted to establish our own PRS in the revised version of the manuscript, in order to eliminate any potential bias through sample overlap. This was done by performing a GWAS on ~105,000 UKB participants who met our QC criteria and have not been exome-sequenced in the employed release. The relevant part of the Methods section now reads:

“In order to establish a PRS model, a GWAS of imputed genotyping data from the UKB was performed. Individuals with no exome sequencing data were selected to ensure no sample overlap, and filtered using the criteria described previously, resulting in 105,565 unrelated (both within this sample and with the 72,024 individuals of the continuous, all- and two-as-control models) male individuals. The imputed genotype data were quality-controlled (info score > 0.3, per-variant missing rate < 5%, HWE $P > 10^{-6}$) and filtered for common variants (MAF > 1%). The GWAS was performed in PLINK 2.0 using the *glm* function, and corrected for age and 18 top PCs (estimated as the optimal number of top PCs for this sample based on lambda calculation). PRS were calculated for the cohort of 72,024 males using PRSice-2 (v2.3.5)⁵⁵ using autosomal and X-chromosomal SNPs $P < 7.85 \times 10^{-3}$ (best-fit PRS P-value threshold) and otherwise default settings.”

Previous work has shown an enrichment of X-chromosomal associations for MPB and the two lead associations discovered in this work also reside in the X chromosome. The authors should include discussion related to the relevance of the X chromosome in MBP in the light of their findings.

We thank the reviewer for this suggestion. We have added the following paragraph to the Discussion section:

“The X-chromosome has long been at the center of genetic analyses on MPHL. Early studies focused on the X-linked androgen receptor gene (*AR*), due to the strict androgen dependency of the phenotype. Although the results have been conflicting in regards to the likely causal variants and genes, the *AR/EDA2R* locus has consistently been the most strongly associated genomic region for MPHL, although neither the precise causal variants nor the causal genes have been confirmed⁴⁰. In the present study, we identified significant associations with four X-chromosomal genes, namely *EDA2R*, *HEPH*, *YIPF6* (Yip1 Domain Family Member 6) and *OPHN1* (Oligophrenin 1), thereby yielding new or additional evidence for these candidate genes. Our analyses did not identify significant associations of rare coding

variants in the *AR* gene ($P_{\text{SKAT-O continuous}} = 3.0 \times 10^{-5}$). This is in line with previous Sanger-sequencing-based studies of the *AR* coding sequence, which did not identify any significant associations between the *AR* and MPHL^{16,41}. Though we cannot exclude the possibility that our analysis lacked statistical power to detect such an association, one might also speculate that a potential involvement of the *AR* gene in MPHL pathobiology is rather impacted by regulatory common variants, than rare coding variants.”

The lack of any associations for AR is fairly surprising, e.g., given that AR missense burden shows an association with testosterone (genebase.org). How many variants were included in the burden tests for AR? How large an effect would have been detected in the 72k men, e.g., with 80% power?

The reviewer is correct in recognizing that the androgen receptor gene is a plausible candidate gene for MPHL due to the androgen dependency of the phenotype and its association with testosterone levels. The *AR/EDA2R* locus has further consistently been the most strongly associated genomic region for MPHL. However, a number of studies have focused on the role of the *AR* in MPHL, and have identified no coding MPHL-associated variants (e.g. through Sanger sequencing of *AR* coding regions (unpublished in-house data; Cobb et al., PLoS One, 2009)) or even provided evidence against the association of previously implicated coding triplet repeats (Ellis et al., Human Genetics, 2007). Based on these prior studies, we do not consider the lack of a (significant) association with *AR* necessarily surprising.

Different studies have further shown that the colocalization of rare and common variant signal is not a trivial task, both because of population structure issues but also because of the presence of potentially distinct underlying biological mechanisms. For instance, several works revealed an enrichment of eQTL regulatory regions in GWAS signal in non-coding regions. The burden or rare impact variants can instead be associated with a larger functional impact on the encoded protein and thus also subject to a different evolutionary pressure. Therefore, there is a large heterogeneity in the evolutionary rate of different genes (which is in turn related to the presence of functionally relevant mutations in coding gene regions).

We also considered performing a burden test power analysis. However, to our knowledge, the only established approach to compute burden test power is PAGEANT (<https://andrewhaoyu.shinyapps.io/PAGEANT/>). The results of this power analysis are strongly affected by the expected phenotypic variance explained by a gene, which is difficult to estimate. In fact, given the complexity of performing a reliable power calculation for rare variants burden test, to our knowledge, power tests are not typically applied in the field of rare variant analyses (in contrast to common variant-based GWAS, for which well-established power calculation based on clear input variables such as AF, effect size and P-value threshold can be applied). We therefore opted not to provide a power calculation. We believe that the current sample size should be enough at least to detect a strong signal, as different significant genes have been detected. Instead, our results indicate a rather moderate signal for the burden of rare variants in the *AR* gene for male baldness, at least in the context of the applied experimental design and the analyzed UK Biobank release.

The second paragraph of the discussion section is overly long and hence exhaustive to follow. Consider cutting text or splitting it into several paragraphs.

We thank the reviewer this remark. We have shortened and split the text into different paragraphs to hopefully improve readability.

Line 164 mentions expression in hair follicles. Which threshold was used to determine expression in this tissue type? Are the genes specific in their expression in hair follicles? How many genes expressed in hair follicles would one expect by chance, i.e., is there any enrichment among the selected genes?

The hair follicle expression data set is based on expression microarray data from 98 human hair follicles (Herrera-Rivero et al., *BMC Dermatol*, 2020). Probes with a detection p-value <0.01 in at least 5% of samples were considered expressed, resulting in 13,217 expressed genes. The overlap between MPHL-associated genes identified in our study and hair follicle-expressed genes (27 / 45) does not constitute an enrichment ($P = 0.35$). An inspection of GTEx bulk sequencing data reveals that a majority of these genes show ubiquitous expression, with only the gene *TCHH* being expressed relatively specifically in hair follicles and skin.

Supplementary tables for gene-based tests identify the genes using gene symbols only. I recommend adding, e.g., ENSEMBL IDs, to avoid potential Excel conversion issues.

We thank the reviewer for this suggestion. In addition to gene symbols, we have now added Ensembl IDs to the supplementary tables and supplementary data.

Reviewer #2:

1. The current results are based on 200k exomes yet UK Biobank have made the full release of 450k exomes available through the Research Analysis Platform. These have been available for a while and would substantially increase the sample size for these analyses. Have the authors looked at that data?

We agree with the reviewer that the analysis of the now available data set of 450k exomes from the UK Biobank would be very desirable in the future. However, such an extended analysis would require a new project application to the UK Biobank as well as working within a different technical structure, which will take several months.

The data presented here are the result of a project that has been ongoing for 2 years in the context and timeline of a PhD project and represent the first detailed analyses of the contribution of rare variants to MPHL. We are convinced that our findings are of importance to the field and fill a gap in the existing literature that justify publication of the data in its present version. To acknowledge the availability of a larger data set, we added the following comment to our conclusion:

“While the present study provides promising first insights into the contribution of rare (coding) variants to MPHL pathobiology based on a tranche of 200,629 exomes from the UK Biobank, the final release of ~450,000 exomes has been released while completing the present analyses. This data set represents a considerable increase in sample size. Continued investigation on the role of rare coding variants for MPHL using this larger data set is therefore warranted.”

2. Gene-based analyses were performed using two methods, SKAT-O and GenRisk. I note that one of the authors has developed GenRisk, but it is not entirely clear why both methods were used, what the differences are, and how they lead to such different results. Given the sample sizes, the results from GenRisk seem to be inflated, did the authors look at inflation estimates (lambda) for both methods? SKAT is a well-established method for gene-based analyses and it is possible to include weights in the analysis, did the authors compare results using a comparable weighting scheme in SKAT? Other than the weights, were the variants included in both analyses the same? I think it would be very important to thoroughly quality-control the results as the majority of follow-up analyses are based on the 124 significantly associated genes (FDR < 0.05) from the GenRisk results.

We thank the reviewer for this comment. The inflation estimates for the GenRisk association analyses based on genes scores derived using internal allele frequency indeed showed moderate inflation for the 3 initially included phenotype definitions (all-model = 1.118, two-as-control-model = 1.194, extreme model = 1.08). However, if rare variants are upweighted using gnomAD allele frequency, no inflation is observed (all-model = 1.0, two-as-control model = 0.987, extreme model = 0.96). Notably, using internal allele frequencies but adding a variant-filter based on biological function (e.g., non-synonymous variants as performed in SKAT-O analysis) also results in lower inflation estimates (all-model = 0.94, two-as-control model = 1.09).

We think that using internal allele frequency could inflate the signal caused by rare high impact variants by taking into account additional rare variants in LD (as no pruning is performed). However, given the relatively large sample size, the inflation could also represent a true polygenic signal for the phenotype. In line with this hypothesis, different GWAS on male baldness also showed moderate inflation (e.g., lambda = 1.15 in Pirastu et al., *Nat Commun*, 2017). It is also worth mentioning that the inflation could be the result of polygenicity of the trait and effects of population structure. To check for artificial inflation, we calculated the lambda1000, in which the calculation of the lambda is scaled down for an equivalent study of 1000 cases and 1000 controls, as suggested and used by multiple studies (PMID: 16228001, 31740837, 33510477). The lambda1000 values for the all-, two-as-control and extreme models were 1.016, 1.004 and 1.017, respectively.

Unfortunately, to the best of our knowledge, in the field of rare variant analysis there are currently no standard approaches to discern between polygenicity and inflation due to LD structure, as can be

performed for common variants analysis by comparing the lambda of the GWAS and the intercept of LD score regression analysis.

As population confounding is more problematic for rare variants analysis, we have now included a sentence in the results section of the main manuscript highlighting that the GenRisk associations based on internal allele frequencies and without any biological filter show moderate inflation. This inflation would be reduced by using external gnomAD allele frequencies or by applying a functional filter on variants to be included in the gene scores. Nevertheless, in order to have a direct comparison with the global SKAT-O analysis, GenRisk association results based on internal allele frequency are shown.

3. For example, the top association for GenRisk is YIPF6, did the authors look at what is driving that association? Interestingly it is not mentioned in the manuscript.

We thank the reviewer for this comment. We dissected the single variant weights in *YIPF6* and observed a larger variability of the GenRisk gene scores when including both allele frequency and CADD, compared to scores based only on AF. This suggests that CADD annotation is introducing individual gene score variability, which contributes to case/control difference (see figure below, which shows variance in scores when including and excluding CADD, scores rescaled between (0,1) for better visualization).

By looking at the case/control ratio (see figure below) and the weight of each *YIPF6* variant based on AF and CADD, 3 variants can be prioritized (score > 20 and case/control ratio >3).

We have added a paragraph in the revised version of the manuscript pointing out the most relevant variants in driving *YIPF6* case/control gene score differences: “*YIPF6*, the most significant gene association in our GenRisk analyses, appears to be driven largely by 3 variants: 23:68521350:T:C (rs989891694, GenRisk-score = 44.9, case/control ratio = 4.0, intronic variant), 23:68498996:A:C (rs2079029887, GenRisk-score = 24.1, case/control ratio = 3.75, upstream variant), and 23:68531882:C:T (rs751994338, GenRisk-score = 20.8, case/control ratio = 5.0, splice region variant).”

4. Does the Conditional GWAS-GenRisk analysis suggest that the GenRisk signal for genes where the GenRisk score attenuates the GWAS signal, are being driven by the GWAS variants? (or variants in linkage disequilibrium).

We thank the reviewer for this comment. The dissection of rare and common variants effect in male baldness was indeed a main focus of our work. By performing conditional analyses between GWAS signal and gene-scores based on rare variants, we aimed at correcting for potential dependency between the common and rare variants signal (i.e., correlation due to LD). Unfortunately, we cannot infer directionality, that is to hypothesize if the GWAS associations are potentially driven by rare variant effects with high impact (e.g., protein-altering variants in coding regions) in LD with GWAS loci, or if the signal is driven by regulatory effects of common variants (e.g., eQTL effects) and the presence of LD with different rare variants leads to a significant gene-burden association. The main aim of the conditional analyses was to perform a gene-prioritization of GWAS signal in gene-dense regions. We observed that in different loci characterized by a large GWAS signal overlapping several genes, the gene-based burden scores show heterogeneity in the associations, suggesting that part of the GWAS signal colocalizes with rare variants in LD in the region nearby to the potentially true associated gene (e.g., possibly the most strongly associated gene).

5. The variants identified in the single variant analyses and included in the conditional analyses (Table 2) have $D^2=1$ (in LDlink), which is consistent with a haplotype effect reported in Hillmer et al., 2009: <https://doi.org/10.1007/s00439-009-0668-z>. This would make the conditional analyses much harder to interpret.

The reviewer is correct in recognizing that the significant variant associations are $D^2 = 1$. When looking at the different haplotype constellations within the UK Biobank (see figure below), we indeed observe that 1) the rare MPH risk allele of the 23:66604439:G:A variant occurs exclusively on the common MPH risk haplotype (rs2497935-A, rs962458-A, rs12007229-C, rs12396249-G) and 2) the rare protective allele of the 23:66197712:C:T variant occurs almost exclusively on a haplotype with only the rs962458-A risk allele (see Figure below). It is therefore possible that the signal of the 23:66197712:C:T variant was due to its occurrence on a lower-risk haplotype, generally confirming the conditional

analysis. As the 23:66604439:G:A variant exclusively occurs in combination with the MPHL risk alleles of the reported risk haplotype, yet a residual signal remains in the conditional analyses, it is possible that this variant exerts an effect independently of the other variants on the risk haplotype. However, at this point we can neither confirm nor exclude a causal role of either variant.

	MPHL risk allele	Haplotype constellations (MPHL risk alleles are marked blue)															
		C	C	C	C	C	T	C	C	C	C	C	T	T	C	C	
23:66197712:C:T	C	C	C	C	C	C	T	C	C	C	C	C	T	T	C	C	
23:66604439:G:A	A	G	G	G	G	A	G	G	G	G	G	G	G	G	G	G	
rs2497935	A	A	G	G	G	A	G	A	G	A	A	G	A	G	G	A	
rs962458	A	A	G	A	A	A	A	G	G	A	A	A	A	G	A	G	
rs12007229	C	C	C	A	C	C	A	C	C	C	A	A	C	C	C	C	
rs12396249	G	G	A	A	G	G	A	A	G	A	A	G	G	A	A	G	
N		58,994	5,454	4,783	1,711	392	130	64	63	28	26	15	3	1	1	1	
Frequency		0.82344	0.07613	0.06676	0.02388	0.00547	0.00181	0.00089	0.00088	0.00039	0.00036	0.00021	0.00004	0.00001	0.00001	0.00001	

We have added this in the conditional single-variant association analysis section of the Discussion in the revised version of the manuscript:

“The two genome-wide significant single variant associations 23:66604439:G:A (in *EDA2R*) and 23:66197712:C:T (in *HEPH*) did not retain genome-wide significance after conditioning, pointing to a (partial) inter-dependence between these variants and common GWAS variants, which was more pronounced for 23:66197712:C:T, while 23:66604439:G:A retained a partial signal. We further observed that i) the rare MPHL risk allele of the 23:66604439:G:A variant occurs exclusively on the common MPHL risk haplotype previously reported by Hillmer et al.³⁹ (rs2497935-A, rs962458-A, rs12007229-C, rs12396249-G) and ii) the rare protective allele of the 23:66197712:C:T variant occurs almost exclusively on a lower-risk haplotype with only the rs962458-A risk allele. While the 23:66604439:G:A variant exclusively occurs on the previously reported MPHL risk haplotype, a partial signal remains in the conditional analyses, which may point to an independent effect of the rare variant and the risk haplotype. However, at this point, a causal role of either variant can neither be confirmed nor excluded.”

6. Did the authors check which other phenotypes highlighted genes were associated with in the public portals such as azphewas.com or genebass.org?

We thank the reviewer for this suggestion. We had not originally checked this, but have included a short paragraph covering previously reported gene associations of these portals in the Discussion section in the revised version of the manuscript:

“Rare coding variants in the associated genes identified in this study have been previously associated with phenotypes and anthropometric indices such as alcohol consumption (*OPHN1*), heel bone mineral density (*LGR4*), bone disorders (*EXT2*), height (*B3GNT8*, *CRHR1*), urea (*HEPH*), and HDL cholesterol (*NR1H3*)^{41,42}. Suggestive associations have further been identified between testosterone levels and *RSPO2* and *EDA2R*. Several of these associations may present interesting links – for instance, epidemiological studies have (albeit with conflicting evidence) found associations between MPHL and alcohol consumption and cardiovascular disease^{43,44}, and genetic correlation analyses have previously identified a genetic overlap of MPHL with height and heel bone mineral density^{13,15}.”

7. Did analyses include a covariate for sequencing batch? (50k vs. 150k)

We thank the reviewer for this comment. We had not previously corrected for sequencing batch, but have now included this covariate in all association analyses of exome sequencing data and added a mention to the Methods section in the revised version of the manuscript:

“The association analyses of the continuous model, all-model <and the two-as-control model were corrected for age, sequencing batch and the top 14 PCs. In the association analyses of the extreme model, correction was made for sequencing batch and the top five PCs only.”

8. Authors should also add a reference for the exomes, Szustakowski et al, (2021): <https://doi.org/10.1038/s41588-021-00885-0>

We thank the reviewer for this suggestion. We have added the reference to Szustakowski et al. (2021) to our initial mentions of the UK Biobank WES data set in the Introduction and Methods sections.

9. Figure 1. HWD is should be HWE (which is also consistent with the text, lines 91/92).

We thank the reviewer for this comment. We have changed HWD to HWE in Figure 1 and its caption in the revised version of the manuscript.

10. It would be helpful to include the rsids for variants in addition to the chromosome:position:alleles names.

We thank the reviewer for this suggestion. We have added rsids for all variants mentioned in the manuscript main text.

11. Typo, line 226: Figure 99

We thank the reviewer for catching this typo, which we have corrected in the revised version of the manuscript.

REVIEWER COMMENTS

Reviewer #1 (Remarks to the Author):

The manuscript has improved in revision. While I am mostly satisfied with the responses to my comments I still ask for additional clarity and validation for the GenRisk results.

A great majority of the discovered gene associations (43/45) result from the GenRisk analysis and the two significant genes from SKAT-O are not among these GenRisk genes. As GenRisk is a new method with few published examples of its use and validation I do consider it pivotal that the details and potential pitfalls of the method are conveyed to the readers in an appropriate manner. For instance, the argument that GenRisk "can be particularly powerful in detecting deleterious gene associations" (line 138) remains elusive. While the authors have now included some details of the comparisons between the output of the two methods (assessment of overlap between GenRisk and SKAT-O genes at a liberal P-value threshold using Fisher's test in the discussion) the differences in the identified genes are still substantial. Could the authors at least examine and elaborate the possible reasons as to why the lead genes in the X chromosome MPH1 locus are completely different between SKAT-O (HEPH and EDA2R, these genes also identified in single variant associations and seem to have plausible functional links to the phenotype) and GenRisk (YIPF6 (and OPHN1))?

Reviewer #2 (Remarks to the Author):

Thank you for your responses.

The abstract states that the aim of the manuscript is "To determine the contribution of rare coding variants to MPH1 etiology". However, reading the responses, it is now clearer that GenRisk includes non-coding variants and as such the differences between SKAT and GenRisk are likely due to the variant selection. For example, for YIPF6, the three variants mentioned as driving the signal are all non-coding.

More importantly, checking those variant three variants for YIPF6 in Ensembl/OpenTargets Genetics, they are all common in Non-Finnish Europeans (MAF \geq 7%), and strongly associated with hair/balding pattern based on publicly available summary statistics. Should these variants not have been filtered out given the MAF<1% filter?

Even if only variants with $MAF < 1\%$ were included, conditional analysis would not necessarily abolish the signal given differences in allele frequency between the variants included in the GenRisk score and the lead GWAS variants.

Therefore, the authors need to check the frequency of the variants included in the GenRisk scores, but in its current form the manuscript is slightly misleading, as it is not only focussed on rare coding variation, but also non-coding variation which is ultimately likely to be tagging the GWAS signals (supported by the GWAS-GenRisk analysis).

REVIEWER COMMENTS

Reviewer #1 (Remarks to the Author):

The manuscript has improved in revision. While I am mostly satisfied with the responses to my comments I still ask for additional clarity and validation for the GenRisk results.

A great majority of the discovered gene associations (43/45) result from the GenRisk analysis and the two significant genes from SKAT-O are not among these GenRisk genes. As GenRisk is a new method with few published examples of its use and validation I do consider it pivotal that the details and potential pitfalls of the method are conveyed to the readers in an appropriate manner. For instance, the argument that GenRisk "can be particularly powerful in detecting deleterious gene associations" (line 138) remains elusive. While the authors have now included some details of the comparisons between the output of the two methods (assessment of overlap between GenRisk and SKAT-O genes at a liberal P-value threshold using Fisher's test in the discussion) the differences in the identified genes are still substantial. Could the authors at least examine and elaborate the possible reasons as to why the lead genes in the X chromosome MPHL locus are completely different between SKAT-O (HEPH and EDA2R, these genes also identified in single variant associations and seem to have plausible functional links to the phenotype) and GenRisk (YIPF6 (and OPHN1))?

We thank the reviewer for these comments, and understand the concerns. As mentioned above, by critical revision of all files and scripts used for the GenRisk analyses, we have indeed discovered an error in the allele frequency annotation in our GenRisk input files, which resulted in the inadvertent inclusion of some variants with MAF>1% and an incorrect variant weight calculation. We have now corrected this error and rerun all GenRisk analyses with the corrected input data. As a result, we no longer observe an association with *YIPF6* and *OPHN1*, but an association with *EDA2R* is observed in both the SKAT-O and GenRisk analyses. A detailed description of the corrected results of the GenRisk analyses can be found in the results and discussion section of the amended manuscript (highlighted in yellow, p.6 lines 135ff and p.10 lines 187ff).

While the overlap between both analyses is more substantial after the correction of the GenRisk input (Fisher's exact test for enrichment of nominally significant ($P < 0.05$) associations between SKATO and GenRisk: $OR_{\text{continuous}} = 8.1$, $P_{\text{continuous}} = 6.2 \times 10^{-123}$, some differences still remain (e.g. an association with *HEPH* is only observed in the SKAT-O analyses). These differences can be explained by differences in the method and filtering scheme between the SKAT-O and GenRisk association tests. Fundamentally, GenRisk is a burden test, while SKAT-O is a hybrid method incorporating both burden and SKAT, which can detect associations with mixed effect directions. GenRisk upweights variants based on functional annotation (in our analyses, CADD score) and MAF; whereas we applied SKAT-O with only upweighting by MAF. The inclusion of a CADD weighting scheme for GenRisk may capture an additional polygenic component and allowed the inclusion of non-coding variation, while stringent filters for e.g., nonsynonymous or even predicted loss-of-function variants are commonly recommended for gene-based association tests.

In order to better understand the impact of these differences, we re-ran GenRisk using the same variants as used in the SKAT-O analysis (nonsynonymous variants in protein-coding genes), while keeping the GenRisk weighting scheme. The top associations of this analysis largely overlap with the top associations also observed in the SKAT-O and unfiltered GenRisk analysis (e.g. *EDA2R*, *EIF3F*, *MITF*, *WNT10A*, see Manhattan plot of the continuous model below), although most genes show attenuated signal.

In these analyses, *HEPH* still showed no notable signal ($P_{\text{continuous}} = 0.91$), suggesting that the inclusion of CADD score in the variant weighting scheme or the underlying method (burden vs hybrid test) are responsible for this difference. The results obtained were similar upon re-running this GenRisk analysis without CADD score weighting ($P_{\text{continuous}} = 0.86$), thereby supporting that this difference arises due to the method, e.g. due to mixed effect directions at the variant-level. The single-variant association test results of the nonsynonymous *HEPH* variants are shown below (beta coefficient <0 shown in blue, beta coefficient >0 shown in red, data based on the continuous model). While our conditional analyses showed that the top (protective) variant strongly drives the *HEPH* gene association, other variants appeared to contribute to this association as well. As the remaining variants show mixed effect directions, the inability of burden tests such as GenRisk to account for mixed effect directions is a plausible explanation for this difference.

This GenRisk analysis filtered for coding variants also yielded an additional significant association in the two-as-control model, namely *EIF3F*. This may indicate that some coding, high impact variants in *EIF3F* contribute strongly to this signal, whereas this signal is slightly diluted in the unfiltered GenRisk analysis due to non-coding variants, and in the SKAT-O analysis due to lower impact coding variants. We have therefore opted to include this filtered GenRisk analysis in the revised version of the manuscript, as it may offer an additional perspective.

In summary, it appears that each of these methods offers advantages: the SKAT-O analysis can capture associations in the presence of mixed effect directions, the GenRisk analysis can capture additional signal through the inclusion of non-coding variants and the upweighting by functional effect, while a GenRisk analysis with filtering of coding variants offers an increased focus on coding high-impact variants, without severe reduction of variant numbers as in the 'high impact' variant filtered SKAT-O analysis.

We have changed the Discussion section highlighting the differences between the methods to read as follows:

“In this study, we performed two types of gene-based analyses: SKAT-O and GenRisk. SKAT-O is a well-established tool for gene-based association analyses and has the ability to detect associations in the presence of mixed effect directions at the variant level. GenRisk employs a scoring system that uses beta distribution weighting schema for allele frequency, which is similar to SKAT-O, and pathogenicity scores (CADD score), to upweight rare and deleterious variants. As a result, GenRisk does not require variant consequence filtering. Moreover, GenRisk generates individual-level gene-based scores, which can be used in downstream analyses such as association analyses and risk prediction modeling. GenRisk was recently used to identify associations between rare genetic variants and blood biomarkers, identifying both known and novel associations (preprint)⁴⁹. In the present study, both methods yielded partially distinct gene associations. While the inclusion of non-coding variants and non-protein-coding genes in the GenRisk analysis may yield overall more comprehensive results, the association signal may encompass a greater overlap with GWAS. The GenRisk analysis of coding variants only, on the other hand, offers an increased focus on high-impact coding variants, without severely reducing the number of variants through e.g., high-impact variant consequence filters. The analyses employed in this study therefore address different hypotheses. While each method offers different biological insights, some identified gene associations are consistent between SKAT-O and GenRisk, and Fisher’s exact tests show a significant overlap of nominally significant associations ($P < 0.05$) between the two analyses across all phenotype models ($OR_{\text{continuous}} = 8.1$, $P_{\text{continuous}} = 6.2 \times 10^{-123}$; $OR_{\text{all-model}} = 9.5$, $P_{\text{all-model}} = 4.4 \times 10^{-134}$; $OR_{\text{two-as-control}} = 9.6$, $P_{\text{two-as-control}} = 7.1 \times 10^{-133}$; $OR_{\text{extreme model}} = 8.8$, $P_{\text{extreme model}} = 7.8 \times 10^{-114}$). However, given the novelty of the approach, corroboration of the GenRisk results in further studies is desirable.”

Furthermore, we have changed the sentence in line 138 (“...can be particularly powerful in detecting deleterious gene associations.”), which now reads:

“To assess the cumulative contribution of rare variants to MPHL, we performed gene-based association analyses using SKAT-O and GenRisk, a new burden association test which upweights rarer and more deleterious variants (based on CADD).”

Reviewer #2 (Remarks to the Author):

Thank you for your responses.

The abstract states that the aim of the manuscript is "To determine the contribution of rare coding variants to MPHL etiology". However, reading the responses, it is now clearer that GenRisk includes non-coding variants and as such the differences between SKAT and GenRisk are likely due to the variant selection. For example, for YIPF6, the three variants mentioned as driving the signal are all non-coding.

More importantly, checking those variant three variants for YIPF6 in Ensembl/OpenTargets Genetics, they are all common in Non-Finnish Europeans (MAF $\geq 7\%$), and strongly associated with hair/balding pattern based on publicly available summary statistics. Should these variants not have been filtered out given the MAF $<1\%$ filter?

Even if only variants with MAF $<1\%$ were included, conditional analysis would not necessarily abolish the signal given differences in allele frequency between the variants included in the GenRisk score and the lead GWAS variants.

Therefore, the authors need to check the frequency of the variants included in the

GenRisk scores, but in its current form the manuscript is slightly misleading, as it is not only focussed on rare coding variation, but also non-coding variation which is ultimately likely to be tagging the GWAS signals (supported by the GWAS-GenRisk analysis).

We thank the reviewer for these insightful comments. As mentioned at the beginning of this document, upon rechecking our inputs and analyses, we noticed discrepancies in the allele frequency annotation of the GenRisk analyses. We apologize for this oversight, and have made the necessary corrections in the revised version of the manuscript. A detailed description of the corrected results of the GenRisk analyses can be found in the results and discussion section of the amended manuscript (highlighted in yellow, p.6 lines 135ff and p.10 lines187ff).

The observation that the inclusion of non-coding variants is likely to tag the GWAS signals is a valuable point, which we have tried to address in the discussion concerning differences between SKAT-O and GenRisk. We have now adapted the relevant section of this paragraph to read:

“In the present study, both methods yielded partially distinct gene associations. While the inclusion of non-coding variants and non-protein-coding genes in the GenRisk analysis may yield overall more comprehensive results, the association signal may encompass a greater overlap with GWAS. The GenRisk analysis of coding variants only, on the other hand, offers an increased focus on high-impact coding variants, without severely reducing the number of variants through e.g., high-impact variant consequence filters. The analyses employed in this study therefore address different hypotheses.”

We agree with the reviewer that the stated focus on rare *coding* variants is misleading and that stating a focus on rare variants, rather than rare coding variants, is more accurate due to the inclusion of non-coding variants in intronic and coding sequence flanking regions in the GenRisk analyses. We have changed the wording accordingly throughout the manuscript.

REVIEWERS' COMMENTS:

Reviewer #1 (Remarks to the Author):

I thank the authors for their thorough reply and the re-assessment of their pipeline and results. The results of the manuscript have changed substantially in the revision, from 45 reported genes to 5. While possibly less exciting and surprising, the findings are now numerically and biologically more convincing. With these changes, the concerns regarding GenRisk are also alleviated to a good degree, yet, I am still somewhat cautious regarding the added value of the method. However, I believe the strengths and weaknesses of the two used gene-based methods are now sufficiently described in the manuscript.

At this point I only suggest two small clarifications to the manuscript:

- Discuss reasons why CEPT1 comes up quite specifically in the primary GenRisk analysis for the all model. Is this possibly related to the model (e.g. phenotype), method or variants chosen?
- Explain the rationale of using several different more liberal p-value thresholds for the enrichment tests and how these thresholds were derived. E.g. $p < 0.05$ used for comparison between GenRisk and SKAT-O, $p < 3e-3$ for GWAS lead SNPs and $p < 3e-4$ for ectodermal dysplasia genes

Reviewer #2 (Remarks to the Author):

The manuscript is much clearer and the authors have addressed all my comments.

REVIEWER COMMENTS

Reviewer #1 (Remarks to the Author):

I thank the authors for their thorough reply and the re-assessment of their pipeline and results. The results of the manuscript have changed substantially in the revision, from 45 reported genes to 5. While possibly less exciting and surprising, the findings are now numerically and biologically more convincing. With these changes, the concerns regarding GenRisk are also alleviated to a good degree, yet, I am still somewhat cautious regarding the added value of the method. However, I believe the strengths and weaknesses of the two used gene-based methods are now sufficiently described in the manuscript.

At this point I only suggest two small clarifications to the manuscript:

- Discuss reasons why *CEPT1* comes up quite specifically in the primary GenRisk analysis for the all model. Is this possibly related to the model (e.g. phenotype), method or variants chosen?

We thank the reviewer for the thorough evaluation of our manuscript, and these helpful suggestions.

In order to better understand what might be driving the association of *CEPT1*, we took another look at the data. The presence of an association in the GenRisk but not the GenRisk filtered analysis (see Table below), suggests that both coding and non-coding variants contribute to this signal.

	P GenRisk	P GenRisk filtered	P SKAT-O
Continuous model	6.47e-03	4.40e-02	6.56e-03
All-model	2.14e-06	5.59e-04	2.50e-04
Two-as-control model	2.12e-02	0.245	1.50e-02
Extreme model	1.76e-02	2.78e-02	3.34e-02

We compared *CEPT1* GenRisk gene scores between cases and controls with and without CADD score weighting, and found that cases and controls only differ when including CADD. We also observe an effect of the phenotype model definition, as the *CEPT1* GenRisk gene scores in pattern groups 2 and 3 are highest on average ($\text{mean}_{\text{pattern1}} = 0.118$, $\text{mean}_{\text{pattern2}} = 0.120$, $\text{mean}_{\text{pattern3}} = 0.120$, $\text{mean}_{\text{pattern4}} = 0.119$). In the continuous model, the lower scores in pattern group 4 may partly counter the association signal, in the two-as-control-model, the effect is present in both the control and the case groups, and in the extreme model, these pattern groups are excluded. However, it is not clear whether this may reflect a contribution of *CEPT1* to specific aspects of the phenotype. We therefore hypothesize that the association of *CEPT1* is driven by both coding and non-coding variants with high CADD scores, as well as the all-model phenotype definition. We have added the following sentence to the results section (lines 131ff):

“The *CEPT1* association finding is likely attributable to a combination of coding and non-coding variants with high CADD scores, and mainly driven by the MPHL pattern groups 2 and 3. Whether this reflects a biological aspect has to be determined by further analyses.”

- Explain the rationale of using several different more liberal p-value thresholds for the enrichment tests and how these thresholds were derived. E.g. $p < 0.05$ used for comparison between GenRisk and SKAT-O, $p < 3e-3$ for GWAS lead SNPs and $p < 3e-4$ for ectodermal dysplasia genes

The selection of the more liberal p-value thresholds was, in essence, arbitrary. We aimed to select a p-value threshold yielding a gene list of ~500 genes for gene set enrichment analyses. This was the case for a p-value threshold of $P < 2.5 \times 10^{-3}$, which we rounded to $P < 3 \times 10^{-3}$. We then adjusted this threshold slightly for the STRING analysis in order to obtain a visually more digestible network of <100 genes, which was obtained with a threshold of $p < 3 \times 10^{-4}$ (resulting in 85 genes).

We agree with the reviewer that the additional p-value threshold of $P < 0.05$ for the comparison of GenRisk and SKAT-O is confusing. We had originally chosen this threshold to show that an enrichment is observed even at this very liberal p-value threshold. In order to improve consistency, we have changed the employed p-value threshold to $P < 3 \times 10^{-3}$ for the GenRisk – SKAT-O comparison. The revised text (lines 360ff) now reads:

“Fisher’s exact tests show a significant overlap of a less stringent set of associations ($P < 3 \times 10^{-3}$) between the two analyses across all phenotype models ($OR_{\text{continuous}} = 54.3$, $P_{\text{continuous}} = 1.5 \times 10^{-17}$; $OR_{\text{all-model}} = 92.8$, $P_{\text{all-model}} = 2.1 \times 10^{-24}$; $OR_{\text{two-as-control}} = 71.3$, $P_{\text{two-as-control}} = 3.5 \times 10^{-20}$; $OR_{\text{extreme model}} = 99.1$, $P_{\text{extreme model}} = 3.9 \times 10^{-19}$).“

We have further added brief clarifications on the choice of the liberal P-value thresholds to the relevant methods paragraphs:

“To improve the feasibility of enrichment analyses and obtain a more comprehensive gene list of approximately 500 genes, a less stringent p-value threshold of $p < 3 \times 10^{-3}$ was selected, resulting in a less stringent set of 595 MPHL-associated genes.” (lines 520ff)

“To further obtain an overview of protein interactions and co-expression, a STRING (v11.5)⁵⁶ protein network analysis was performed using a less stringent set of MPHL-associated genes. Here, a threshold of $P < 3 \times 10^{-4}$ was selected in order to obtain a more manageable network of < 100 genes.“ (lines 568ff)

Reviewer #2 (Remarks to the Author):

The manuscript is much clearer and the authors have addressed all my comments.

We once again thank the reviewer for the careful evaluation of our manuscript. We are happy that all comments and concerns could be addressed with our previous response.